# SAP: Exact Sorting in Splatting via Screen-Aligned Primitives

**Zhanke Wang[1], Zhiyan Wang[1], Kaiqiang Xiong[1,2], Jiahao Wu[1,2], Yang Deng[1], Ronggang Wang[1,2]***

[1]Guangdong Provincial Key Laboratory of Ultra High Definition Immersive Media Technology,
Shenzhen Graduate School, Peking University, [2]Peng Cheng Laboratory
{zk_wang, zywang23, xiongkaiqiang, wjh0616, dengyang}@stu.pku.edu.cn
rgwang@pkusz.edu.cn

## Abstract

Recently, 3D Gaussian Splatting (3DGS) has achieved state-of-the-art rendering results. However, its efficiency relies on simplifications that disregard the thickness of Gaussian primitives and their overlapping interactions. These simplifications can lead to popping artifacts due to inaccurate sorting, thereby affecting the rendering quality. In this paper, we propose Screen-Aligned Primitives (SAP), an anisotropic kernel that generates primitives parallel to the image plane for each view. Our rasterization pipeline enables full per-pixel ordering in real time. Since the primitives are parallel for a given viewpoint, a single global sorting operation suffices for correct per-pixel depth ordering. We formulate 3D reconstruction as a combination of a 3D-consistent decoder and 2D view-specific primitives, and further propose a highly efficient decoder to ensure 3D consistency. Moreover, within our framework, the primitive function values remain consistent between view space and screen space, allowing arbitrary radial basis functions (RBFs) to represent the scene without introducing projection errors. Experiments on diverse datasets demonstrate that our method achieves state-of-the-art rendering quality while maintaining real-time performance.

## 1 Introduction

Neural Radiance Fields (NeRF) (1; 2; 3; 4; 5) have demonstrated exceptional performance in 3D scene representation and novel view synthesis, but come with high computational costs. Recently, 3D Gaussian Splatting (3DGS) (6) has garnered widespread attention by introducing the rasterization of primitive points, achieving real-time rendering while maintaining rendering quality. Building on this foundation, researchers have explored various representations, including 2D planar Gaussians (7), generalized Gaussian kernels (8), and neural Gaussians (9; 10). However, these methods rely on the coarse sorting mechanism inherent to 3D Gaussian Splatting, which fundamentally limits their depth ordering precision.

A critical limitation of 3DGS (6) is *popping artifact*, which arises from its depth sorting mechanism. Specifically, 3DGS performs depth sorting based solely on Gaussian centers before alpha blending, assuming that Gaussians do not overlap. For anisotropic ellipsoids, sorting based solely on centers is inherently inaccurate in practice. As illustrated in Fig. 1(c), while Gau1 appears before Gau2 in view-space depth ordering, certain pixels require Gau2 to be rendered first due to overlapping regions. Several methods have been proposed to address this issue. StopThePop (11) employs the maximum of a 1D Gaussian along the view ray as a more precise depth estimation. However, it also neglects Gaussian overlap and lacks the capability to perform full per-pixel sorting, as shown in Fig. 1(d). EVER (12) employs ray tracing for the per-pixel rendering accuracy, but at the cost of doubling the

---

*Corresponding author

39th Conference on Neural Information Processing Systems (NeurIPS 2025).

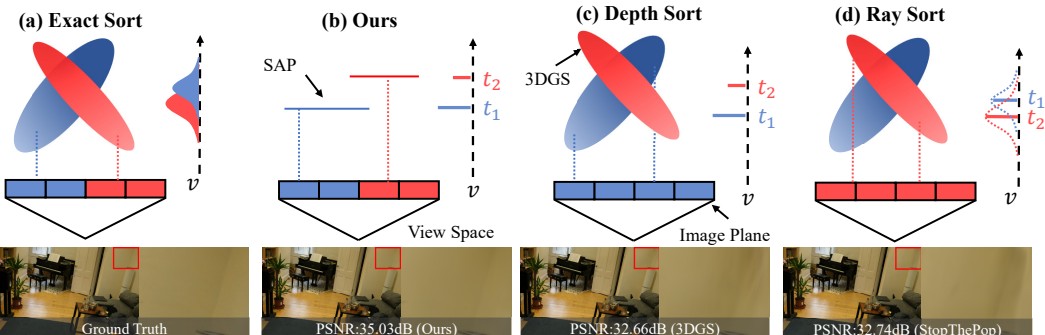

Figure 1: **Comparative analysis of sorting strategies:** (a) Precise per-pixel sorting ensures the correct rendering order for every individual pixel. (b) Our Screen-Aligned Primitives guarantee accurate per-pixel ordering through a single sorting pass, inherently eliminating overlap artifacts. (c) View-space depth ordering (6) prioritizes Gau1 ■ over Gau2 ■ across all overlapping pixels when Gaussians intersect. (d) Sorting by maximum pulse intensity of 1D Gaussians (11) places Gau2 first, yet fails to distinguish per-pixel coverage.

rendering overhead. In this work, we improve the original representation of 3DGS by constraining the primitives to be parallel to the image plane. To fundamentally eliminate overlaps between different primitives, our primitives are parallel thin slices, ensuring that they do not intersect. Under this condition, depth sorting along the z-axis becomes equivalent to exact per-pixel sorting, enabling accurate ordering without introducing additional computational overhead. Another drawback of 3DGS is that it uses an affine projection approximated by the Jacobian matrix (13) to construct the 2D Gaussian plane in the screen space, which introduces projection errors. Our parallel primitives eliminate this projection error because there is no variation in depth within the same primitive. We can perform exact projection through simple geometric scaling, while enabling exploration of more flexible and varied kernel representations.

If we directly define an explicit representation, it is impossible to determine a primitive that remains parallel to the image plane across all viewing directions. By leveraging anchor-based Gaussians (10; 9; 14), we introduce neural networks to generate primitives aligned parallel to each specific plane. This design ensures that our Screen-Aligned Primitives maintain strict parallelism with the image plane across all perspectives. We design a 3D-consistent decoder based on the FiLM architecture (15), which is widely used in controllable image generation (16). Unlike conventional usage, we condition the decoder on the viewing direction to learn a direction-aware 3D representation. In summary, our framework employs a 3D-consistent decoder to model view-dependent representations while utilizing 2D primitives for each viewpoint. Since our representation is anisotropic, the conventional densification strategy may fail under large viewpoint variations. To address this, we introduce a maximum positional gradient-based densification method to improve reconstruction quality.

In summary, the main contributions of our paper are as follows:

- We introduce a 3D-consistent decoder which enables our framework to be constructed on a 3D feature network combined with 2D reconstruction primitives.
- We propose Screen-Aligned Primitives (SAP) representation that ensures more accurate sorting.
- We demonstrate that our framework supports accurate and unbiased projection, allowing for the use of more flexible and expressive kernel representations.
- We present a densification strategy based on the maximum positional gradient, which increases point density in regions with significant view-dependent variation, thereby enhancing reconstruction quality.

## 2  Related Work

**Novel View Synthesis.**  NVS has emerged as a pivotal task in the fields of computer vision and graphics, with significant implications for applications such as media generation (17), virtual reality

(18), and autonomous driving. Over the past few years, NeRF (1) has demonstrated remarkable performance in this field, attracting widespread attention. NeRF employs a multilayer perceptron (MLP) to model scenes and utilizes ray marching for image rendering. However, NeRF requires multiple queries to the neural network for each pixel, leading to considerable computational overhead. Although various approaches (19; 20; 21; 22; 23) have been proposed to accelerate the speed of NeRF, they continue to face notable limitations.

**Gaussian Splatting.** 3D Gaussian Splatting (3DGS) (6) represents scenes as collections of anisotropic Gaussian ellipsoids , achieving photorealistic rendering quality and real-time performance through an efficient tile-based rasterizer. It relies on the splatting method (13) to construct 2D Gaussians from 3D Gaussians. Subsequent research rapidly expanded the capabilities of 3DGS across various tasks, such as anti-aliasing (24; 25), 3D generative modeling (26; 27; 28), dynamic modeling (29; 30; 31; 32), surface reconstruction (33; 7; 34; 35), and densification improvement (36; 37; 38; 39). These methods optimize 3DGS from various perspectives, allowing it to adapt to different applications.

At the same time, some works are dedicated to exploring better scene representation methods. Among them, several (7; 40; 41) focus on modifying the kernel expressions of the functions to address more complex scenes and tasks. Although these methods offer more flexible representations, they all adopt the original sorting approach of 3DGS. We propose a new framework in which arbitrary kernels can be used without considering projection errors. Additionally, we extend the Gaussian kernel (8) in our framework to achieve a more compact representation.

**Popping Artifact and Sorting.** The original work on splatting was presented by (42). Popping artifacts persist as an intrinsic limitation of such projective rendering paradigms. 3D Gaussian Splatting (6) employs EWA splatting (13) to project anisotropic 3D Gaussians onto the imaging plane. (43) introduced aligning sheet buffers parallel to the image plane, and (44; 45) introduced slice-based volume rendering methods that do not suffer from popping artifacts. Although 2DGS (7) uses Gaussian planes to represent the scene, it still does not address the intersection issues between different primitives. Unlike traditional methods based on explicit graphical primitives, we leverage deep learning to fit a rendering primitive that is parallel to any viewpoint.

Recently, several methods based on 3DGS (6) have been dedicated to mitigate *popping artifact*. StopThePop (11) uses the maximum value of a 1D Gaussian as depth, improving popping artifacts during view rotation. 3DGRT (46)and EVER (12) accurately determine the intersection points between primitives and rays by ray tracing. Sort-Free (47) calculates alpha blending using order-independent transparency and (48) approximates the integration of values along the ray to calculate transmittance more accurately. However, these methods still involve many approximations in their computations, and ray-tracing-based approaches incur significantly higher computational cost. In contrast, our method achieves outstanding results while retaining the original fast rasterization pipeline.

## 3 Preliminaries

### 3.1 3D Gaussian Splatting

3D Gaussian splatting represents scenes as a set of differentiable semi-transparent particles defined by their kernel function and renders images by rasterizing the projected 2D counterparts. Each 3D Gaussian $\mathcal{G}^{3D}$ centered at a point $\mu \in \mathbb{R}^3$ with covariance matrix $\Sigma \in \mathbb{R}^{3 \times 3}$ is given by:

$$\mathcal{G}^{3D}(x) = e^{-\frac{1}{2}(x-\mu)^T \Sigma^{-1}(x-\mu)}, \Sigma = RSS^T R^T, \tag{1}$$

where $x \in \mathbb{R}^3$ is arbitrary position within the scene, $R \in \mathbb{R}^{3 \times 3}$ is a rotation and $S \in \mathbb{R}^{3 \times 3}$ a scaling matrix.

The view-dependent appearance of each 3D Gaussian is modeled by third-order spherical harmonics (SH) coefficients $F \in \mathbb{R}^{3 \times 16}$, combined with opacity $\sigma \in \mathbb{R}$. To render an image, we need to project our 3D Gaussian $\mathcal{G}^{3D}$ to 2D as a 2D Gaussian $\mathcal{G}^{2D}$ for rendering. The 2D covariance matrix $\Sigma'$ is given as follows:

$$\Sigma' = JW\Sigma W^T J^T, \tag{2}$$

where $W$ is a viewing transformation and $J$ is the Jacobian of the affine approximation of the projective transformation. The tile-based rasterizer employs depth $t \in \mathbb{R}$ sorting in viewspace to render 3D Gaussian primitives and employs $\alpha$-blending to compute the color of pixel $x'$.

## 3.2 Anchor-Based Gaussian Splatting

To obtain primitives parallel to the plane from different viewpoints, we employ anchor-based Gaussian splatting (9; 10) to characterize Gaussian attributes. Each $k$ anchor carries a position coordinate $x_v \in \mathbb{R}^3$, a local feature $\hat{\mathbf{f}}_a \in \mathbb{R}^{32}$, and $l_a \in \mathbb{R}^3$ is a scaling factor controlling the predicted offsets $\mathcal{O} \in \mathbb{R}^{k \times 3}$. The positions $\mu$ of neural Gaussians are calculated as:

$$\{\mu_0, ..., \mu_{k-1}\} = x_v + \{\mathcal{O}_0, ...\mathcal{O}_{k-1}\} \cdot l_v. \tag{3}$$

Additionally, a corresponding tiny MLP $F$ decodes the opacities, scales, rotations, and colors from the anchor features, as well as the distance between the anchor and the camera $\delta_{vc}$ and the viewing direction $\mathbf{d}_{cv}$.

From anchor-based Gaussian Splatting, we derive a key observation: if entirely distinct primitives are generated for each view, and each viewpoint only observes a single face of the 3D Gaussians, then the necessity of maintaining a full 3D representation becomes questionable. This insight motivates our exploration of 2D primitives as an alternative.

## 4 Methodology

To achieve exact sorting during primitive rendering, we first introduce a 3D-consistent decoder that generates planar primitives for each viewing direction (Sec. 4.1). Based on this decoder, we further propose a Screen-Aligned Primitives (SAP) framework to address Gaussian sorting and overlap issues without incurring additional time overhead (Sec. 4.2). Leveraging the SAP framework, we enable unbiased projection, enabling the use of flexible kernel functions to represent the scene more accurately (Sec. 4.3). Finally, we detail the training procedure and densification strategy (Sec. 4.4).

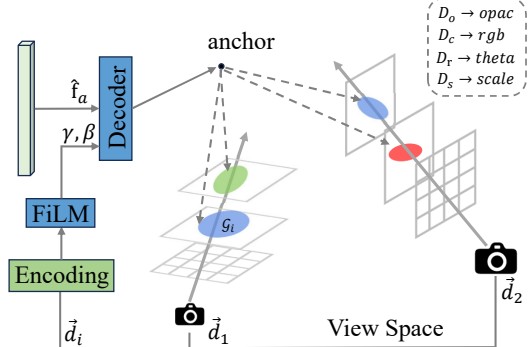

Figure 2: **Schematic illustration of SAP.** For each input viewpoint, we generate screen-aligned primitives directly in view space using anchor-based decoding. A lightweight decoder takes the anchor features and a directional encoding of the viewing direction as input, and outputs 2D primitives aligned with the corresponding view plane. Unlike prior methods that typically operate in world space, our approach generates primitives that are consistently aligned with the viewing direction.

### 4.1 3D-Consistent Decoder

We decompose the 3D reconstruction process into a 3D-consistent decoder and a 2D primitive-based renderer. In this section, we present the design and formulation of our 3D-consistent decoder. In NeRF (1), researchers typically employ positional encoding combined with viewing directions as input to an MLP for color and density prediction. Subsequent works such as Instant-NGP (19) improve upon this by introducing hash-based encoding, while PlenOctree (23) incorporates spherical harmonics to assist directional learning. Similarly, in Gaussian representations based on anchors (9; 10), local anchor features concatenated with viewing directions are fed into a lightweight MLP for decoding.

**Directional encoding.** In this paper, we employ a direction decoding framework based on the FiLM network (15), where the direction serves as a control signal, as shown in Fig. 2. Additionally, we explore the impact of different directional encodings on the results. Our experiments reveal that using spherical harmonic direction encoding significantly improves the decoding and rendering quality. We employ *real-valued spherical harmonics* (SH) to encode the input unit direction vector $\vec{d} = (x, y, z)$. The spherical harmonics encoding up to degree $l$ are defined as:

$$\text{SH}(\vec{d}) = \left[ Y_0^0(\vec{d}), Y_1^{-1}(\vec{d}), \dots, Y_l^l(\vec{d}) \right],$$

where $Y_l^m(\cdot)$ denotes the basis function of the real spherical harmonics of degree $l$ and order $m$, evaluated in direction $\vec{d}$.

**Direction-modulated MLP.** We use a Direction-modulated MLP as the 3D decoder $\mathcal{D}$. Given an input feature $\hat{\mathbf{f}}_a$ and a viewing direction $\mathbf{d}$, our network first transforms the input via a linear layer:

$$\mathbf{h}_1 = W_1 \hat{\mathbf{f}}_a + \mathbf{b}_1, \tag{4}$$

then applies feature-wise linear modulation using scale and shift parameters derived from the direction encoding $\text{SH}(\vec{d})$:

$$\boldsymbol{\gamma}, \boldsymbol{\beta} = \text{MLP}_{\text{dir}}(\text{SH}(\vec{d})), \quad \mathbf{h}_2 = \phi(\boldsymbol{\gamma} \odot \mathbf{h}_1 + \boldsymbol{\beta}), \tag{5}$$

where $\phi(\cdot)$ is a non-linear activation, and $\odot$ denotes element-wise multiplication. Finally, the output is predicted via a second linear layer:

$$\theta, \mathbf{s} = \text{sigmoid}(W_2 \mathbf{h}_2 + \mathbf{b}_2). \tag{6}$$

where $\theta$ and $\mathbf{s}$ denote the rotation and anisotropic scale parameters, respectively, and $\text{sigmoid}(\cdot)$ is the element-wise sigmoid activation function. For the learning of other parameters, we follow the settings of Scaffold-GS (9).

## 4.2 Screen-Aligned Primitives

3DGS (6) performs a global sort of Gaussians based on the view-space z-coordinate $t_i = \mu_i^{(z)}$ of their mean. However, since each 3D Gaussian is defined by a covariance matrix $\Sigma_i$, which defines an anisotropic ellipsoid in a 3D space. When Gaussians overlap, their per-pixel depth orders may contradict the global sorting, resulting in rendering artifacts such as incorrect transparency blending.

We construct a plane parallel to the screen through the center of the viewpoint space, thereby constraining all primitives to be parallel. As shown in Fig. 2, for each specific viewpoint, our rendering primitives are all parallel. This planar alignment enables single-pass global sorting to guarantee pixel-accurate depth ordering, as all primitives share identical orientation relative to the view direction. Within this parallel plane, each primitive is parameterized by 2D rotation $\theta \in [0, 2\pi)$ and scaling $\mathbf{s} = (s_x, s_y)$. We do not perform 3D rotations; instead, we construct 2D transformations within the local plane. We explicitly construct the view-space covariance matrix $\Sigma_v$ through a composition of 2D scaling matrix $S_v \in \mathbb{R}^{2 \times 2}$ and rotation matrix $R_v \in \mathbb{R}^{2 \times 2}$, as detailed below:

$$\Sigma_v = R_v S_v S_v^T R_v^T. \tag{7}$$

For a given viewpoint, we effectively use 2D primitives to reconstruct the 2D image. In this case, our rendering method is similar to (49), where a 2D Gaussian function is used to reconstruct the 2D image.

In summary, our framework employs a 3D-consistent decoder to generate view-specific 2D shape parameters for Screen-Aligned Primitives, which are then sorted by depth $t$.

## 4.3 Unbiased Projection

**Screen-Aligned Primitives Splatting.** Gaussian Splatting (6) projects 3D ellipsoids onto the imaging plane as 2D Gaussians. To preserve the differentiability, it employs an affine approximation (13) constructed via the Jacobian matrix, Eq. 2. However, this approximation introduces geometric distortions, particularly under perspective projection with significant depth variations or strong parallax effects. The affine projection considers only the depth at the center, making it accurate only near the center point, with increasing error as the distance from the center grows. Some works (50; 51; 52; 53) have analyzed this projection error and introduced more accurate projection computations.

Since our parallel primitives exhibit no depth variation, their projection onto the imaging plane simplifies to a geometric scaling centered at the original depth $t$:

$$P = \begin{bmatrix} \frac{f_x}{t} & 0 \\ 0 & \frac{f_y}{t} \end{bmatrix}, \tag{8}$$

where $f_x, f_y$ are the horizontal and vertical focal lengths of the camera, respectively. Therefore, our screen-space covariance matrix $\Sigma_s$ is constructed as $\Sigma_s = P\Sigma_v P^T$.

In fact, the view-space Mahalanobis distance and screen-space Mahalanobis distance in our framework are equivalent, as proven in Appendix B.5. Specifically:

$$(x_v - \mu_v)^T \Sigma_v^{-1}(x_v - \mu_v) = (x_s - \mu_s)^T \Sigma_s^{-1}(x_s - \mu_s). \tag{9}$$

where $(\cdot)_v$ and $(\cdot)_s$ represent variables in the view coordinate system and the screen coordinate system, respectively. Therefore, the distributions in view space $\mathcal{G}_v$ and screen space $\mathcal{G}_s$ are equivalent under our parameterization, i.e.,

$$\mathcal{G}_v(\mu_v, \Sigma_v) \equiv \mathcal{G}_s(\mu_s, \Sigma_s). \tag{10}$$

**Versatile Kernel Representation.** The edge distribution of a 3D Gaussian is still Gaussian, which simplifies the computation of splatting. Ges (8) utilizes a generalized Gaussian kernel, but due to projection errors, it approximates the kernel by introducing a scaling parameter in place of the exact generalized Gaussian kernel function. However, in our framework, the distribution in the view space and the screen space are equivalent, allowing us to define any kernel. Formally, let $\mathcal{K}(\cdot)$ be an arbitrary differentiable kernel function; our projective isomorphism guarantees:

$$\mathcal{K}_v(x_v; \mu_v, \Sigma_v) \equiv \mathcal{K}_s(x_s; \mu_s, \Sigma_s). \tag{11}$$

Building upon this equivalence, we generalize the Gaussian kernels. Let the Mahalanobis distance be defined as $M(x; \mu, \Sigma) = (x - \mu)^T \Sigma^{-1}(x - \mu)$ and $p \in (0, \infty)$, our generalized kernel function is expressed as:

$$\mathcal{K}^{Ges}(x; \mu, \Sigma) = e^{-\frac{1}{2}M^p(x;\mu,\Sigma)}, \quad M(x; \mu, \Sigma) \in [0, \infty), \tag{12}$$

The Generalized Gaussian kernel $\mathcal{K}^{Ges}$ defines more flexible primitives, allowing us to achieve similar rendering quality with fewer primitives. At the same time, we restrict the value range of the generalized Gaussian kernel $\mathcal{K}^{Ges}$ to within a $3^{1/p}$ long axis range, with a detailed proof available in Appendix B.5.

## 4.4 Optimization

**Improved Densification.** Similar to prior methods (6; 9), we utilize positional gradients as a criterion for dynamic anchor growth. However, previous approaches used averaging positional gradients across all training views. This uniform averaging fails to prioritize underoptimized viewpoints with high gradient variance, leading to insufficient anchor growth in regions that are poorly reconstructed from specific perspectives. To address this, we introduce max-gradient $\nabla_{\max}$ that prioritizes the most significant positional gradients across viewpoints $\mathcal{V}$:

$$\|\nabla_{\max}\| = \max_{k \in \mathcal{V}} \|\frac{\partial \mathcal{L}_k}{\partial \mu_{\mathbf{2D}}}\|. \tag{13}$$

When using average position gradient densification, some regions are prone to gradient vanishing, resulting in holes or blurriness. To balance the robustness of the average gradients and the sensitivity of the maximum gradients, our densification condition combines both metrics. Growth occurs when either the average gradient magnitude exceeds $\tau_1$ or the maximum gradient magnitude exceeds $\tau_2$. $\tau_1$ and $\tau_2$ are the predefined thresholds.

**Loss.** We retain the original loss function of 3DGS (6). As our primitives do not overlap, scaling regularization (9) is not necessary. The loss function integrates an $\mathcal{L}_1$ metric combined with a loss of structural similarity (SSIM) $\mathcal{L}_{ssim}$:

$$\mathcal{L} = \lambda_1 \mathcal{L}_1 + \lambda_{ssim} \mathcal{L}_{ssim} \tag{14}$$

# 5 Experiments

## 5.1 Experimental Setup

**Dataset and Metrics.** We evaluated Screen-Aligned Primitives (SAP) on a diverse set of real-world datasets to demonstrate its superior visual quality. Specifically, we tested our approach on various public datasets, including 9 scenes from MipNeRF360 (3), 2 scenes from Tanks&Temples (54), 2

Table 1: **Quantitative evaluation on the Mip-NeRF360 (3), Tanks&Temples (54), and Deep Blending (55) datasets.** Our method achieves the best rendering quality on most datasets. We follow the experimental settings and dataset partitioning of Scaffold-GS (9).

| Dataset | Tanks&Temples | | | Mip-NeRF 360 | | | Deep Blending | | |
|---|---|---|---|---|---|---|---|---|---|
| Method | PSNR↑ | SSIM↑ | LPIPS↓ | PSNR↑ | SSIM↑ | LPIPS↓ | PSNR↑ | SSIM↑ | LPIPS↓ |
| Mip-NeRF 360 (3) | 22.22 | 0.758 | 0.257 | 27.56 | 0.793 | 0.234 | 29.40 | 0.900 | 0.245 |
| 3D-GS (6) | 23.71 | 0.845 | 0.178 | 27.43 | 0.813 | 0.217 | 29.46 | 0.900 | 0.247 |
| 2DGS (7) | 22.96 | 0.825 | 0.217 | 27.03 | 0.804 | 0.239 | 29.49 | 0.899 | 0.259 |
| Scaffold-GS (9) | 24.19 | 0.854 | 0.174 | 27.55 | 0.810 | 0.232 | 30.28 | 0.909 | 0.239 |
| StopThePop (11) | 23.21 | 0.843 | 0.173 | 27.28 | 0.810 | 0.213 | 29.86 | 0.904 | 0.234 |
| DisC-GS (58) | 24.96 | 0.866 | 0.120 | 28.01 | 0.833 | 0.189 | 30.42 | 0.907 | 0.199 |
| 3DGS-MCMC (39) | 24.29 | 0.860 | 0.190 | 28.00 | 0.831 | 0.178 | 29.67 | 0.890 | 0.320 |
| **SAP (Ours)** | 25.04 | 0.870 | 0.145 | 28.05 | 0.835 | 0.208 | 30.02 | 0.910 | 0.236 |

Table 2: **Ablation on the Tanks&Temples Dataset (54).** We evaluate the performance of two different rendering primitives: 3D Gaussian Primitives (6; 9) and Screen-Aligned Primitives, Sec. 4.2; FiLM: FiLM-modulated MLP, Sec. 4.1; SH encoding: real-valued spherical harmonics(SH) encoding, Sec. 4.1; $\nabla_{max}$: maximum view-direction gradient threshold, Sec. 4.4. The numerical difference from the best-performing result is shown in the bottom-right corner.

| Components | | | 3DGaussian Primitives | | | Screen-Aligned Primitives | | |
|---|---|---|---|---|---|---|---|---|
| FiLM | SH encoding | $\nabla_{max}$ | PSNR ↑ | SSIM ↑ | LPIPS ↓ | PSNR ↑ | SSIM ↑ | LPIPS ↓ |
| | | | $24.15_{-0.89}$ | $0.858_{-0.012}$ | $0.163_{+0.018}$ | $24.59_{-0.45}$ | $0.846_{-0.024}$ | $0.181_{+0.036}$ |
| ✔ | | | $24.25_{-0.79}$ | $0.858_{-0.012}$ | $0.165_{+0.020}$ | $24.67_{-0.37}$ | $0.847_{-0.023}$ | $0.179_{+0.034}$ |
| | ✔ | | $24.45_{-0.59}$ | $0.859_{-0.011}$ | $0.163_{+0.018}$ | $24.79_{-0.25}$ | $0.853_{-0.017}$ | $0.175_{+0.030}$ |
| | | ✔ | $24.30_{-0.74}$ | $0.862_{-0.008}$ | $0.154_{+0.009}$ | $24.68_{-0.36}$ | $0.859_{-0.011}$ | $0.161_{+0.016}$ |
| ✔ | ✔ | | $24.52_{-0.52}$ | $0.860_{-0.010}$ | $0.165_{+0.020}$ | $24.74_{-0.30}$ | $0.852_{-0.018}$ | $0.176_{+0.031}$ |
| ✔ | | ✔ | $24.42_{-0.62}$ | $0.861_{-0.009}$ | $0.156_{+0.011}$ | $24.80_{-0.24}$ | $0.860_{-0.010}$ | $0.161_{+0.016}$ |
| | ✔ | ✔ | $24.51_{-0.53}$ | $0.863_{-0.007}$ | $0.155_{+0.010}$ | $24.89_{-0.15}$ | $0.858_{-0.012}$ | $0.164_{+0.019}$ |
| ✔ | ✔ | ✔ | $24.56_{-0.48}$ | $0.864_{-0.006}$ | $0.155_{+0.010}$ | **25.04** | **0.870** | **0.145** |

scenes from DeepBlending (55). These datasets cover a range of environments, from bounded indoor spaces to unbounded outdoor settings, providing a comprehensive evaluation of the performance of our method. We adopt the experimental settings and dataset partitioning as established in Scaffold-GS (9). In particular, for the MipNeRF 360 dataset (3), scenes with resolutions exceeding 1600 are downsampled to 1600. Consistent with previous studies, we assess reconstruction quality using three metrics: PSNR ↑, SSIM ↑ (56), and LPIPS ↓ (57). We color each cell as best , second best , and third best . We refer to our Screen-Aligned Gaussian kernel as SAP.

**Implementation Details.** Our PyTorch implementation is built upon the Scaffold-GS (9) framework, in which we re-implement the 3D-consistent decoder. We modified the CUDA kernel to replace spatial primitives with screen-aligned primitives. These modifications do not affect the original rendering efficiency; in fact, by reducing matrix computations (from 3D matrices to 2D matrices), our approach achieves a slight speed-up over the baseline in constant time. We retained most of the parameter settings from Scaffold-GS to ensure a fair comparison. The difference is that our parallel kernel leverages the prior along the z-axis. Therefore, instead of using the anchor view direction, we modify the input of the Decoder for covariance to be the camera's z-axis direction. Detailed hyperparameter settings and model architecture are provided in Appendix B.2 and Appendix B.3. We conducted all experiments on a single NVIDIA L40S GPU.

## 5.2 Results and Comparisons

**Quantitative results.** We evaluated SAP against various state-of-the-art techniques in both novel view synthesis tasks. The quantitative results on three datasets (55; 3; 54) are presented in Table 1, with additional details per scene available in Appendix B.9. We sequentially compare MipNeRF360 (3), 3DGS (6), 2DGS (7), Stop-the-Pop (11), Scaffold-GS (9), DisC-GS (58) and 3DGS-MCMC

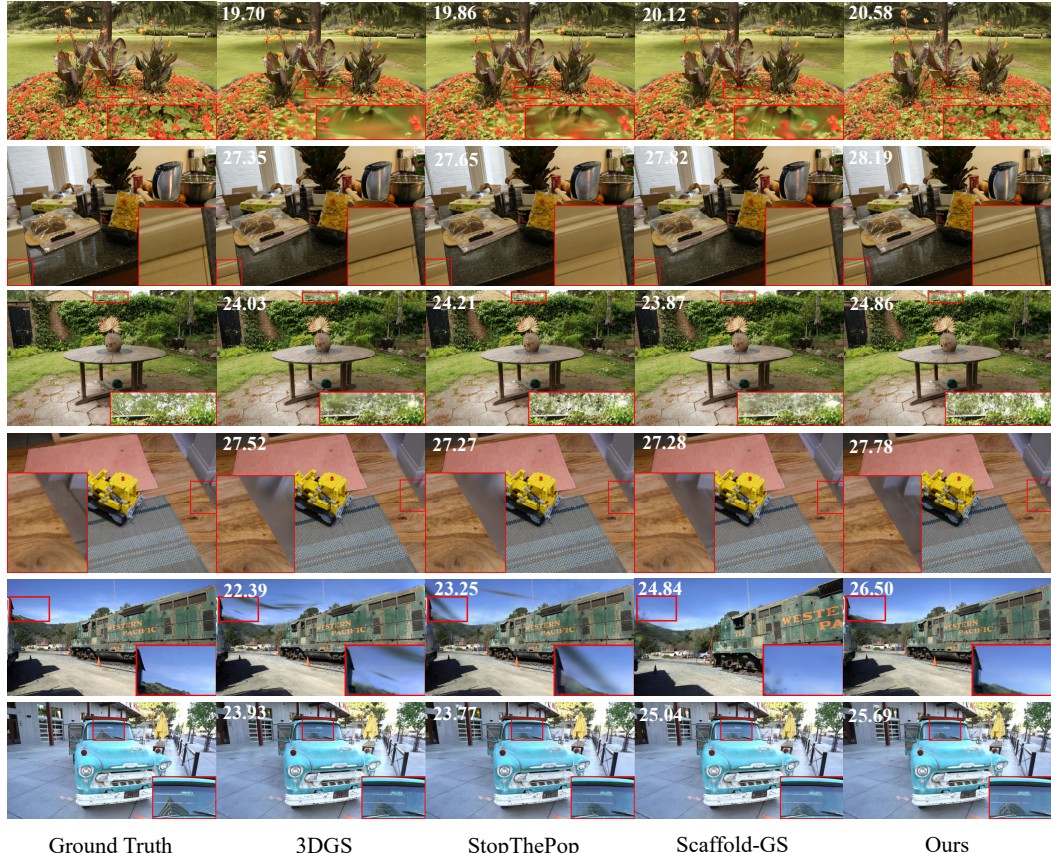

| Ground Truth | 3DGS | StopThePop | Scaffold-GS | Ours |

Figure 3: **Qualitative comparisons across diverse datasets (55; 3; 54).** We compare 3DGS (6), Stop-the-Pop (11), and Scaffold-GS (9) sequentially. Regions where our method outperforms others are highlighted in red. Additionally, the **PSNR** of each image is annotated in the top-left corner. Our approach achieves high-quality reconstruction even in several challenging areas, whereas other methods commonly exhibit blurriness, floating artifacts, and other inconsistencies in these regions. This can be attributed to the accuracy of our parallel primitive ordering and our effective densification strategy.

Table 3: **Comparison of different encodings.** We conduct a comparative evaluation of various encoding methods, namely: no positional encoding, MLP encoding (MLP), Fourier encoding (FE), and spherical harmonics encoding (SH), on the Tanks&Temples Dataset (54).

| no encoding | MLP | FE | SH |
|---|---|---|---|
| PSNR↑ / SSIM↑ / LPIPS↓ | PSNR↑ / SSIM↑ / LPIPS↓ | PSNR↑ / SSIM↑ / LPIPS↓ | PSNR↑ / SSIM↑ / LPIPS↓ |
| 24.74 / 0.852 / 0.176 | 24.80 / 0.856 / 0.169 | 24.88 / 0.863 / 0.161 | **25.04 / 0.870 / 0.145** |

(39). As shown in Table 1, our SAP achieves the best results on Tanks & Temples as well as Mip-NeRF360, ranking first or second in nearly all metrics. The table indicates that although 2DGS (7) improve geometric accuracy, they compromise rendering quality. Our 2D screen-aligned primitive representation, however, not only preserves but also significantly boosts rendering quality, outperforming even the state-of-the-art 3DGS baseline.

**Qualitative results.** In Fig. 3, we showcase the comparisons between our method and other approaches. We can observe that our method significantly reduces artifacts and aliasing compared to previous approaches. For example, in the fifth row, other methods produce artifacts, whereas ours avoids them. This improvement is due to the accurate ordering of our primitives, which prevents interference between different primitives during rendering, ensuring that primitives intended for one region are not mistakenly rendered in another. Additionally, our method reduces blurriness, as

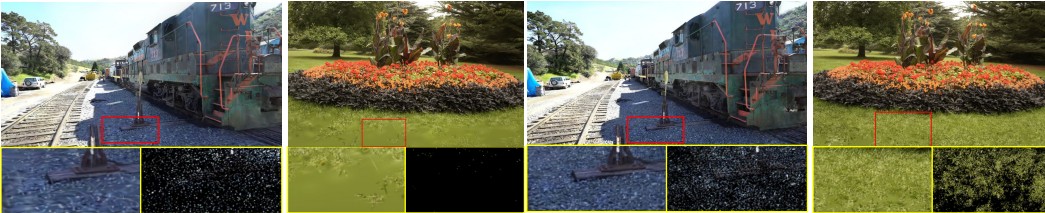

Figure 4: **Comparison of the effects of two splitting methods.** We have magnified the rendered image and the point distribution of the highlighted region. In the left two columns, the original average position gradient is employed, which results in numerous voids leading to blurring; in the right two columns, the combination of average and maximum position gradients effectively fills these voids, yielding improved rendering quality.

Table 4: **Comparison of different kernels.** We compare Anchor-3DGS (9), Anchor-2DGS (10; 7), Ges (8), and our three different kernels, on the Tanks&Temples Dataset (54).

| Scaffold-GS (9) | Anchor 2DGS (7; 10) | Ges (8) | SAP-Epa | SAP-Ges | SAP-Gaussian |
|---|---|---|---|---|---|
| PSNR↑ / SSIM↑ / LPIPS↓ | PSNR↑ / SSIM↑ / LPIPS↓ | PSNR↑ / SSIM↑ / LPIPS↓ | PSNR↑ / SSIM↑ / LPIPS↓ | PSNR↑ / SSIM↑ / LPIPS↓ | PSNR↑ / SSIM↑ / LPIPS↓ |
| 24.19 / 0.854 / 0.174 | 23.52 / 0.835 / 0.199 | 23.35 / 0.836 / 0.198 | 20.53 / 0.667 / 0.409 | 24.75 / 0.861 / 0.166 | **25.04 / 0.870 / 0.145** |

shown in the fourth row of Fig. 3, particularly on the ground. This improvement is attributed to our densification strategy, which effectively fills in the gaps. As illustrated in Fig. 4, we zoomed in on the red-boxed area and separately presented the rendering results and the distribution of primitive points. It can be intuitively observed from the figure that our densification strategy can improve scene rendering.

## 5.3 Ablation Studies

**Efficacy of Screen-Aligned Primitives.** Table. 2 evaluates the impact of various configurations and kernel choices on the rendering quality for the Tanks&Temples dataset. We conduct a comprehensive evaluation of the rendering quality variations resulting from different combinations of our SAP and 3D Gaussian primitives with other system components. Each of our components achieves improvements over the baseline, among which directional encoding and the SAP kernel yield the largest enhancements. Our FiLM decoder also achieves better 3D-consistent decoding performance compared to the MLP decoder. We believe that employing FiLM for directional modulation enables more effective learning of direction-dependent attributes than simply concatenating direction vectors with features as input to an MLP. Furthermore, our maximum densification of the position gradient $\nabla_{max}$ effectively improves the quality of the scene reconstruction. The original densification strategy tends to fail when there is a significant viewpoint difference, whereas maximum position gradient densification alleviates this issue effectively.

**Results of different directional encoding.** We explore how various directional encoding strategies affect the rendering quality in Table 3. In particular, we compare four methods: (1) no directional encoding; (2) a simple MLP that encodes the direction into a 32-dimensional feature vector; (3) Fourier encoding adopted from NeRF (1); and (4) hard-coded spherical harmonics encoding. Our results show that the spherical harmonics encoding yields the best performance. This can be attributed to the inherent design of spherical harmonic functions, which are well-suited for capturing directional properties, making them particularly effective as control signals for enforcing 3D consistency and learning direction-dependent attributes.

**Results of different kernels.** We compared the rendering quality for different kernels in Table 4. Scaffold-GS (9), also known as Anchor-3DGS. In Octree-GS (10) , Anchor-2DGS was implemented by combining sliced-plane Gaussian sheets (7) with anchors. Ges (8) provides a generalized Gaussian kernel. In our comparisons, SAP-Epa adopts the generalized Epanechnikov kernel (59). For clarity, SAP-Ges utilizes a 2D generalized Gaussian kernel (8), as defined in Eq. 12, whereas the standard SAP-Gaussian employs 2D Gaussian primitives. Specifically, we tested SAP-Ges with $p$ as a learnable parameter. SAP-Gaussian is a special case of SAP-Ges when $p = 1$. Our SAP-Gaussian achieves the best reconstruction quality. Our experimental results show that, although Ges is a superset of

Gaussian kernels, it performs worse than the standard Gaussian kernel in rendering quality. We believe this to be likely because Gaussian Splatting serves as a mixture of multiple primitives, and when the number of primitives is sufficient, the specific shape of each kernel has limited impact on rendering quality. Additionally, the higher complexity of the Ges representations complicates the optimization process, making it harder to identify the best learning parameters.

## 6   Limitations and conclusions

**Limitations.**   Although our method explores two different kernel behaviors, both are radial basis functions constructed based on Mahalanobis distance. In fact, within our framework, we can experiment with a wider range of kernels, not limited to radial basis functions, such as asymmetric kernels (60; 40). Furthermore, since our approach does not take surface normals (7) into account, it may present challenges in mesh extraction. This limitation will be addressed in future work.

**Conclusions.**   In this work, we introduce Screen-Aligned Primitives (SAP), a novel framework for 3D representation. Our approach employs a neural network decoder to enforce parallel alignment between each primitive and the rendered plane, enabling accurate primitive ordering at a reduced computational cost. Our framework decomposes the 3D representation into a combination of a 3D decoder and 2D primitive representations. Additionally, we demonstrate that our primitives are adaptable to various kernel functions. Extensive experiments on several challenging datasets validate that SAP achieves real-time rendering performance.

**Acknowledgments.**   This work is financially supported by Guangdong Provincial Key Laboratory of Ultra High Definition Immersive Media Technology(Grant No. 2024B1212010006), this work is also financially supported for Outstanding Talents Training Fund in Shenzhen, Shenzhen Science and Technology Program(Grant No. SYSPG20241211173440004 and Grant No. RCJC20200714114435057).

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

# Appendix

## A  Broader Impacts and Limitation

### A.1  Broader Impacts

Our method focuses on exploring a novel representation for 3D reconstruction and improving the rendering quality of 3D reconstruction. It does not have a direct social impact. However, it may trigger a chain reaction in some downstream applications. For example, when using 3DGS to reconstruct digital humans, if the rendering quality is high enough to make it difficult for people to distinguish between real and fake, it may raise social and ethical concerns, as well as issues related to the forgery of personal portrait assets. Our method can improve the quality of 3D rendering in certain scenarios, which may exacerbate such risks. This is a concern that any technology should take into account, and we urge users to carefully consider the potential consequences when applying our method.

### A.2  Limitation

Our method is applicable to various kernel representations, not limited to Gaussian kernels. In this paper, we only explore the Gaussian kernel and the generalized Gaussian kernel; however, our approach is, in fact, generalizable to the vast majority of kernels. Many existing works have investigated alternative kernel representations (60; 40), many of which are based on 2DGS (7) to leverage its inherent 2D structure and the convenience of projection-based computation. Similarly, our method adopts a 2D representation and is compatible with many of these approaches, which could provide significant inspiration for future research in this field.

Our method focuses on improving the quality of 3D rendering without incorporating normals to enhance the geometric consistency, such as (7; 61). This may cause difficulties in applications like surface reconstruction and mesh extraction.

Our method does not address lighting and physical reflection. Prior works (14) have shown that fitting complex lighting conditions can significantly improve rendering quality. We consider this to be a valuable direction for future research.

## B  More Technical Details

### B.1  Overview

This appendix is organized as follows: (Sec. B.2) Experimental Setup; (Sec. B.3) Network Architecture; (Sec. B.4) Proof of Unbiased Projection; (Sec. B.5) More Details of SAP-Ges; (Sec. B.6) Experiments on Computational Resources; (Sec. B.7) View-dependent Gaussian Attribute Decoding; (Sec. B.8) More Ablation on Kernels; (Sec. B.9) Additional Results;

### B.2  Experimental Setup

**Experimental Setup on the Mip-NeRF360 Dataset.**   It is worth noting that in Scaffold-GS (9), images with resolutions larger than 1600 are downsampled to 1600. In contrast, some methods such as 3DGS-MCMC (39) downsample outdoor scenes by a factor of 4 and indoor scenes by a factor of 2. These differences in preprocessing can lead to slight variations in the results. In this paper, we follow the experimental setup of Scaffold-GS.

**Hyperparameter Settings.**   Most of our hyperparameter settings follow those of Scaffold-GS. Specifically, we used 10 neural Gaussians for each anchor; We set the threshold for the average gradient magnitude ($\tau_1$) to 0.0002, and the threshold for the maximum gradient magnitude ($\tau_2$) to 0.0015; The initial learning rate of our 3D-consistent decoder is set at 0.004 and gradually annealed to 0.00004 during training; the learning rate for the features is set to 0.0025. All other hyperparameter settings follow those of Scaffold-GS to ensure a fair comparison.

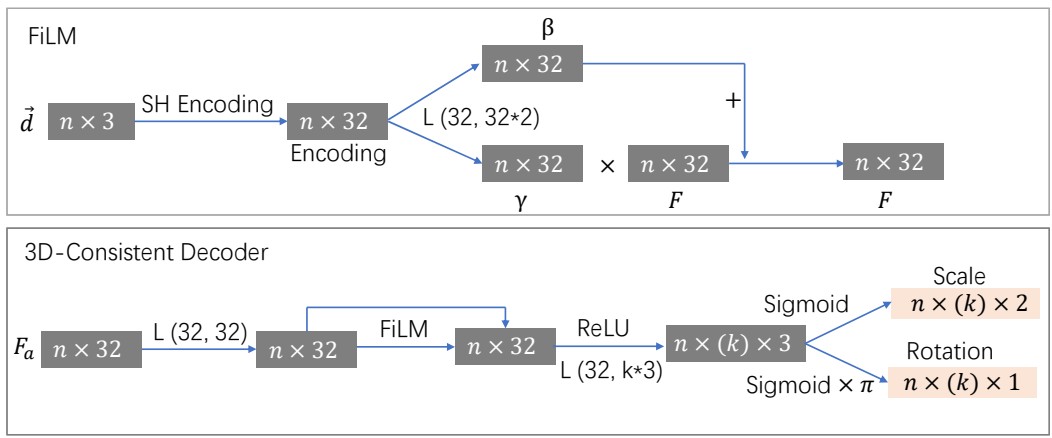

Figure 5: **Illustration of the 3D-Consistent Decoder.** The top shows the direction modulator, while the bottom depicts the full decoder architecture. Here, $L(\cdot)$ denotes a linear layer.

## B.3 Network Architecture

Our 3D-Consistent Decoder architecture is illustrated in Fig. 5. We condition the network on the viewing direction, which is first encoded and passed through a linear layer to generate modulation parameters. These parameters are then used to linearly modulate the features to produce the final output. The rest of our network architecture remains consistent with that of Scaffold-GS (9).

## B.4 Proof of Unbiased Projection

As shown in Fig. 6, 3DGS (6) introduces errors due to the use of an approximate affine projection, whereas our method can effectively avoid this issue.

In our projection framework, the function values of the screen-space kernel and the view-space kernel are equivalent. If we ignore the $z$-axis depth, assuming the view-space mean point is $(\mu_x^v, \mu_y^v)$ and the screen-space mean point is $(\mu_x^s, \mu_y^s)$. For a given depth $t$ and horizontal and vertical focal lengths $f_x = \frac{\text{width}}{2*\text{tanFovx}}, f_y = \frac{\text{height}}{2*\text{tanFovy}}$, the transformation matrix from space to screen, based on our geometric scaling projection, can be expressed as:

$$P = \begin{bmatrix} \frac{f_x}{t} & 0 \\ 0 & \frac{f_y}{t} \end{bmatrix}. \tag{15}$$

The relationship between the covariance matrices $\Sigma^v$ and $\Sigma^s$ can be expressed as:

$$\Sigma^s = P\Sigma^v P^T. \tag{16}$$

For the camera model established in 3DGS, the perspective projection matrix $M_{\text{perspective}}$ can be expressed as:

$$\begin{bmatrix} \frac{2*z_{near}}{right-left} & 0 & \frac{right+left}{right-left} & 0 \\ 0 & \frac{2*z_{near}}{top-bottom} & \frac{top+bottom}{top-bottom} & 0 \\ 0 & 0 & \frac{z_{far}}{z_{far}-z_{near}} & \frac{-z_{far}*z_{near}}{z_{far}-z_{near}} \\ 0 & 0 & 1 & 0 \end{bmatrix}. \tag{17}$$

With the additional conditions: bottom $= -$top; left $= -$right; $z_{\text{near}} = 0.1$; $z_{\text{far}} = 100$ ; top $=$ tanFovx $\cdot z_{\text{near}}$ ; right $=$ tanFovy $\cdot z_{\text{near}}$. We can simplify it to:

$$\begin{bmatrix} \frac{1}{\text{tanFovx}} & 0 & 0 & 0 \\ 0 & \frac{1}{\text{tanFovy}} & 0 & 0 \\ 0 & 0 & 1 & -0.01 \\ 0 & 0 & 1 & 0 \end{bmatrix}. \tag{18}$$

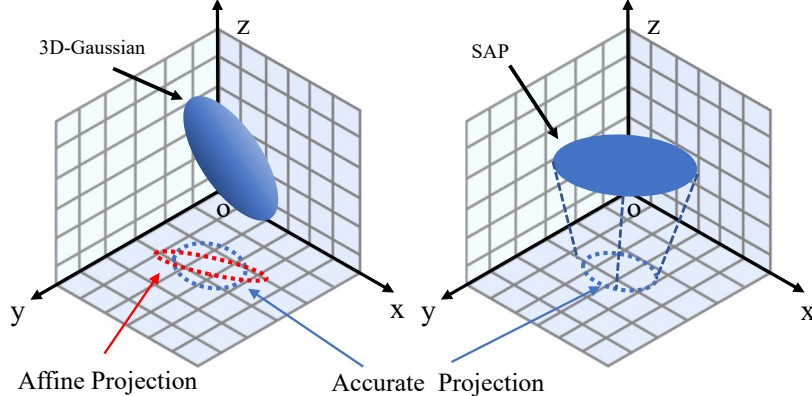

Figure 6: **A simplified diagram of different projection methods.** On the left is the projection process of the 3D Gaussian sphere: the blue represents the standard perspective projection, while the red represents the affine projection approximated by the Jacobian matrix. On the right is our 2D slice transformation, where our projection is an affine transformation that achieves the same effect as the exact projection.

Therefore, for any view-space coordinates $(x^v, y^v, t, 1)$, the transformation to screen space can be written as:

$$
\begin{bmatrix} x^n \\ y^n \\ z^n \\ w^n \end{bmatrix} = M_{\text{perspective}} \begin{bmatrix} x^v \\ y^v \\ t \\ 1 \end{bmatrix} \tag{19}
$$

where $w^n = t$ and $(x^n, y^n, z^b, w^n)$ are the NDC coordinates.. Therefore, the normalized coordinates $x^n$ and $y^n$ in screen space are given by $\frac{x^n}{t}$ and $\frac{y^n}{t}$, respectively. That is,

$$
\begin{bmatrix} x^n \\ y^n \end{bmatrix} = \begin{bmatrix} \frac{x^v}{\text{tanFovx}*t} \\ \frac{x^v}{\text{tanFovx}*t} \end{bmatrix} \tag{20}
$$

According to the defined NDC transformation:

$$
\begin{bmatrix} x^s \\ y^s \end{bmatrix} = \begin{bmatrix} \frac{(x^n+1)*\text{width}-1}{2} \\ \frac{(y^n+1)*\text{height}-1}{2} \end{bmatrix} \tag{21}
$$

Therefore, for any given $(x^s, y^s)$, and $(\mu_x^s, \mu_y^s)$,

$$
\begin{bmatrix} x^s - \mu_x^s \\ y^s - \mu_y^s \end{bmatrix} = \begin{bmatrix} \frac{f_x}{t} * (x^v - \mu_x^v) \\ \frac{f_y}{t} * (y^v - \mu_y^v) \end{bmatrix} \tag{22}
$$

The relationship between them is:

$$
\begin{bmatrix} x^s - \mu_x^s \\ y^s - \mu_y^s \end{bmatrix} = P \begin{bmatrix} x^v - \mu_x^v \\ y^v - \mu_y^v \end{bmatrix} \tag{23}
$$

Finally, the distance function value in screen space is:

$$
\begin{bmatrix} x^s - \mu_x^s \\ y^s - \mu_y^s \end{bmatrix}^T \Sigma^{-s} \begin{bmatrix} x^s - \mu_x^s \\ y^s - \mu_y^s \end{bmatrix} = \begin{bmatrix} x^v - \mu_x^v \\ y^v - \mu_y^v \end{bmatrix}^T P^T P^{-T} \Sigma^{-v} P^{-1} P \begin{bmatrix} x^v - \mu_x^v \\ y^v - \mu_y^v \end{bmatrix}
$$
$$
= \begin{bmatrix} x^v - \mu_x^v \\ y^v - \mu_y^v \end{bmatrix}^T \Sigma^{-v} \begin{bmatrix} x^v - \mu_x^v \\ y^v - \mu_y^v \end{bmatrix} \tag{24}
$$

In summary, in our framework, we can define any functional kernel using the Mahalanobis distance. At this point, the function values in screen space and view space are equivalent. For 3DGS, an affine projection, approximated by the Jacobian matrix, is used to approximate the covariance. Since the projection from 3D to 2D is non-invertible, this identity does not hold.

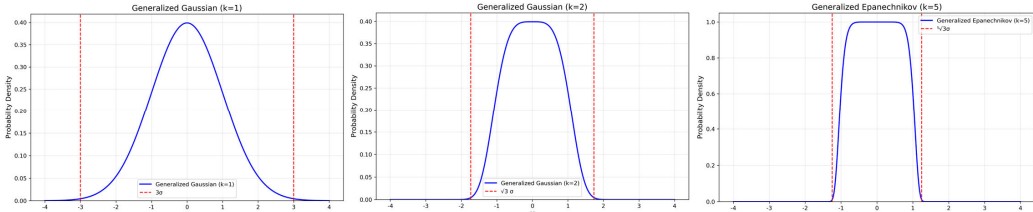

Figure 7: Generalized kernal Function.

Table 5: **Timings.** We compare the training and inference time of our method with that of Scaffold-GS (9), evaluating both with and without our proposed densification strategy, for the "Train" scene in Tanks&Temples dataset (54).

| Train
Metrics \| Method | Ours
wo $\nabla_{max}$ | Ours
w $\nabla_{max}$ | Scaffold-GS (9)
wo $\nabla_{max}$ | Scaffold-GS
w $\nabla_{max}$ |
|---|---|---|---|---|
| PSNR ↑ | 23.56 | 23.76 | 22.77 | 22.89 |
| #Anchor(k) ↓ | 393.8 | 584.4 | 353.3 | 520.0 |
| Training Time ↓ | 26 minutes | 32 minutes | 24 minutes | 29 minutes |
| Rendering FPS ↑ | 80 | 75 | 88 | 79 |

## B.5 More Details of SAP-Ges

In this paper, we extend the generalized Gaussian kernel:

$$\mathcal{K}^{Ges}(x; \mu, \Sigma) = e^{-\frac{1}{2}M^p(x)} \quad M(x) \in [0, \infty). \tag{25}$$

Its illustration is shown in the first row of Fig. 7. For 3DGS, its pixel radius is $3\lambda$, where $\lambda$ is the largest eigenvalue of the covariance matrix, which corresponds to the major axis. To ensure that $(3\lambda, 0)^T \Sigma^{-s}(3\lambda, 0) \leq 9$, 3DGS imposes a condition on the covariance matrix $\Sigma^s$. This condition guarantees that the squared Mahalanobis distance for the point $(3\lambda, 0)$ is less than or equal to 9, ensuring that the distance function does not exceed a certain threshold. To ensure that the distance value is less than 9, we take the radius of the generalized Gaussian kernel as $3^{1/p}\lambda$. The specific formula is:

$$\left( \left(3^{1/p}\lambda, 0\right)^T \Sigma^{-s} \left(3^{1/p}\lambda, 0\right) \right)^p \leq 9. \tag{26}$$

When $p > 1$, the generalized Gaussian kernel is more compact than the Gaussian kernel. When $p < 1$, the radius of the generalized Gaussian kernel becomes larger.

## B.6 Experiments on Computational Resources

Our method shares a rendering pipeline that is highly similar to 3DGS (6), so the training and rendering times are approximately equivalent given the same number of primitives. However, because of potential differences in the gradients produced by our method compared to the original one, different numbers of points may be generated under the corresponding densification strategies. While the number of points is approximately proportional to the computation time, our method demonstrates superior rendering quality under a comparable point count. We report the training and rendering overhead of our method and the baseline under two different densification strategies, as shown in Table 5.

## B.7 View-dependent Gaussian Attribute Decoding

We visualized our Gaussian attribute decoder, as shown in Fig. 8. Specifically, we uniformly sampled directions on the sphere and used these as input to the decoder, obtaining the corresponding decoded attributes for each direction. We visualized properties such as color, opacity, rotation, and scale. As

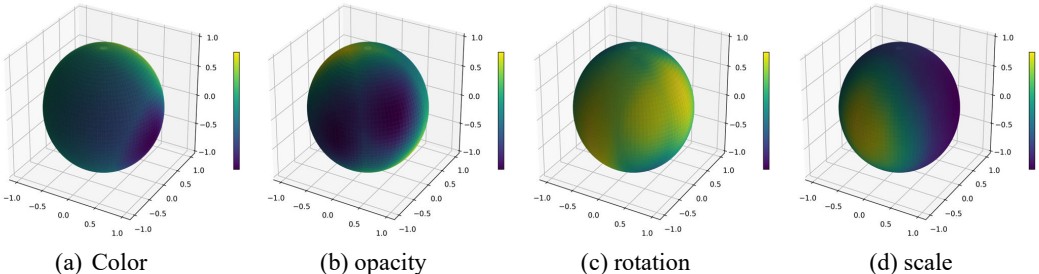

| (a) Color | (b) opacity | (c) rotation | (d) scale |

Figure 8: **View-dependent Gaussian attribute decoding.** We uniformly sample directions on the sphere to evaluate the performance of our attribute decoder.

Table 6: **Comparison of different kernels.** We compared different parameter designs of SAP-Ges.

| SAP-Ges($p = 2$) | SAP-Ges($p = 0.5$) | SAP-Ges($p_{learnable}$) | SAP-Gaussian($p = 1$) |
| --- | --- | --- | --- |
| PSNR↑ / Mem↓ / FPS↑ | PSNR↑ / Mem↓ / FPS↑ | PSNR↑ / Mem↓ / FPS↑ | PSNR↑ / Mem↓ / FPS↑ |
| 24.65 / 124.61 / **90** | 24.43 / **86.83** / 70 | 24.52 / 99.89 / 76 | **25.04** / 103.53 / 82 |

illustrated, our decoder maintains a degree of continuity across adjacent viewpoints while exhibiting anisotropic characteristics.

### B.8 More Ablation on Kernels

We conducted experiments on SAP-Ges with different parameter settings, on the Tanks&Temples Dataset (54), Table 6:

- $p = 2$: This produces a more compact kernel than the Gaussian, resulting in more points and higher memory usage, but faster rendering.
- $p = 0.5$: This leads to a wider kernel range with fewer points and the lowest memory consumption. However, rendering is the slowest because of large Gaussian extents.
- Learnable $p$: The results lie between the above two. We attribute this to the challenge of finding an optimal learning rate for $p$.
- Gaussian kernel ($p = 1$): This setting achieves the best rendering performance with a good trade-off between memory and speed.

### B.9 Additional Results

Here we list the error metrics used in our evaluation across all methods and scenes considered, as shown in Tab. 7 8.

Table 7: Quantitative evaluation of rendering efficiency per scene in Mip-NeRF360(3).

| | bicycle | flowers | garden | stump | treehill | room | counter | kitchen | bonsai | Avg. |
|---|---|---|---|---|---|---|---|---|---|---|
| | | | | | PSNR↑ | | | | | |
| Mip-NeRF360(3) | 24.40 | 21.64 | 26.94 | 26.36 | 22.81 | 31.40 | 29.44 | 32.02 | 33.11 | 27.57 |
| 3D-GS (6) | 25.18 | 21.48 | 27.24 | 26.62 | 22.45 | 31.49 | 28.98 | 31.35 | 32.10 | 27.43 |
| 2D-GS (7) | 24.82 | 20.99 | 26.91 | 26.41 | 22.52 | 30.86 | 28.45 | 30.62 | 31.64 | 27.03 |
| StopThePop (11) | 25.20 | 21.50 | 27.16 | 26.69 | 22.43 | 30.83 | 28.59 | 31.13 | 31.93 | 27.28 |
| Scaffold-GS (9) | 24.81 | 21.42 | 27.17 | 26.27 | 23.08 | 31.93 | 29.34 | 31.30 | 32.70 | 27.55 |
| DisC-GS (58) | - | - | - | - | - | - | - | - | - | 28.01 |
| 3DGS-MCMC (39) | **25.67** | **22.09** | **27.65** | **27.47** | **23.2** | 32.32 | 29.26 | 31.82 | 32.54 | 28.00 |
| **SAP(Ours)** | 25.26 | 21.48 | 27.48 | 26.76 | 23.02 | **32.88** | **29.94** | **31.90** | **33.73** | **28.05** |

| | bicycle | flowers | garden | stump | treehill | room | counter | kitchen | bonsai | Avg. |
|---|---|---|---|---|---|---|---|---|---|---|
| | | | | | SSIM↑ | | | | | |
| Mip-NeRF360 | 0.693 | 0.583 | 0.816 | 0.746 | 0.632 | 0.913 | 0.895 | 0.920 | 0.939 | 0.793 |
| 3D-GS | 0.763 | 0.603 | 0.862 | 0.772 | 0.632 | 0.917 | 0.906 | 0.925 | 0.939 | 0.814 |
| 2D-GS | 0.731 | 0.573 | 0.845 | 0.764 | 0.630 | 0.918 | 0.908 | 0.927 | 0.940 | 0.804 |
| StopThePop | 0.767 | 0.604 | 0.862 | 0.775 | 0.635 | 0.917 | 0.903 | 0.925 | 0.939 | 0.814 |
| Scaffold-GS | 0.725 | 0.587 | 0.842 | 0.784 | 0.644 | 0.925 | 0.914 | 0.928 | 0.946 | 0.810 |
| DisC-GS | - | - | - | - | - | - | - | - | - | 0.833 |
| 3DGS-MCMC | **0.784** | 0.609 | 0.866 | **0.810** | 0.665 | 0.934 | **0.921** | 0.937 | 0.950 | 0.831 |
| **SAP(Ours)** | 0.758 | **0.631** | **0.875** | 0.780 | **0.707** | **0.945** | 0.915 | **0.939** | **0.962** | **0.835** |

| | bicycle | flowers | garden | stump | treehill | room | counter | kitchen | bonsai | Avg. |
|---|---|---|---|---|---|---|---|---|---|---|
| | | | | | LPIPS↓ | | | | | |
| Mip-NeRF360 | 0.289 | 0.345 | 0.164 | 0.254 | 0.338 | 0.211 | 0.203 | 0.126 | 0.177 | 0.234 |
| 3D-GS | 0.213 | 0.338 | 0.109 | 0.216 | 0.327 | 0.221 | 0.202 | 0.127 | 0.206 | 0.217 |
| 2D-GS | 0.271 | 0.378 | 0.138 | 0.263 | 0.369 | 0.214 | 0.197 | 0.125 | 0.194 | 0.239 |
| StopThePop | 0.206 | 0.335 | **0.107** | 0.210 | 0.319 | 0.216 | 0.200 | 0.126 | 0.202 | 0.213 |
| Scaffold-GS | 0.256 | 0.359 | 0.146 | 0.284 | 0.338 | 0.202 | 0.191 | 0.126 | 0.185 | 0.232 |
| DisC-GS | - | - | - | - | - | - | - | - | - | 0.189 |
| 3DGS-MCMC | **0.202** | **0.246** | 0.112 | **0.194** | **0.300** | 0.181 | **0.174** | **0.114** | 0.176 | **0.188** |
| **SAP(Ours)** | 0.227 | 0.314 | 0.123 | 0.248 | 0.305 | **0.173** | 0.192 | 0.123 | **0.169** | 0.208 |

Table 8: Quantitative evaluation of rendering efficiency per scene in Tanks &Temples Dataset (54) and Deep Blending (55) .

| | PSNR↑ | | | | | |
| | Truck | Train | Avg. | Dr Johnson | Playroom | Avg. |
|---|---|---|---|---|---|---|
| Mip-NeRF360 | 24.91 | 19.52 | 22.22 | 29.14 | 29.66 | 29.40 |
| 3D-GS | 25.39 | 22.04 | 23.71 | 29.06 | 29.86 | 29.46 |
| 2D-GS | - | - | 22.96 | - | - | 29.49 |
| StopThePop | 24.93 | 21.48 | 23.21 | 29.42 | 30.31 | 29.86 |
| Scaffold-GS | 25.89 | 22.48 | 24.19 | **29.73** | **30.83** | 30.28 |
| DisC-GS | - | - | 24.96 | - | - | **30.42** |
| 3DGS-MCMC | 26.11 | 22.47 | 24.29 | 29.00 | 30.33 | 29.67 |
| **SAP(Ours)** | **26.31** | **23.76** | **25.04** | 29.62 | 30.41 | 30.02 |

| | SSIM↑ | | | | | |
| | Truck | Train | Avg. | Dr Johnson | Playroom | Avg. |
|---|---|---|---|---|---|---|
| Mip-NeRF360 | 0.857 | 0.660 | 0.758 | 0.901 | 0.900 | 0.900 |
| 3D-GS | 0.878 | 0.813 | 0.845 | 0.898 | 0.901 | 0.900 |
| 2D-GS | - | - | 0.825 | - | - | 0.899 |
| StopThePop | 0.878 | 0.808 | 0.843 | 0.903 | 0.905 | 0.904 |
| Scaffold-GS | 0.885 | 0.822 | 0.854 | **0.910** | 0.907 | 0.909 |
| DisC-GS | - | - | 0.866 | - | - | 0.907 |
| 3DGS-MCMC | 0.890 | 0.830 | 0.860 | 0.890 | 0.900 | 0.895 |
| **SAP(Ours)** | **0.905** | **0.835** | **0.870** | 0.908 | **0.911** | **0.910** |

| | LPIPS↓ | | | | | |
| | Truck | Train | Avg. | Dr Johnson | Playroom | Avg. |
|---|---|---|---|---|---|---|
| Mip-NeRF360 | 0.159 | 0.354 | 0.257 | 0.237 | 0.252 | 0.245 |
| 3D-GS | 0.148 | 0.208 | 0.189 | 0.247 | 0.246 | 0.247 |
| 2D-GS | - | - | 0.217 | - | - | 0.259 |
| StopThePop | 0.142 | 0.204 | 0.173 | 0.234 | **0.235** | 0.234 |
| Scaffold-GS | 0.143 | 0.204 | 0.174 | 0.235 | 0.242 | 0.239 |
| DisC-GS | - | - | **0.120** | - | - | **0.199** |
| 3DGS-MCMC | 0.140 | 0.240 | 0.190 | 0.330 | 0.310 | 0.320 |
| **SAP(Ours)** | **0.114** | **0.175** | 0.145 | **0.228** | 0.244 | 0.236 |

