# OpenReview forum: "SAP: Exact Sorting in Splatting via Screen-Aligned Primitives"
_NeurIPS.cc/2025/Conference — NeurIPS 2025 poster_

### Official Review · Reviewer_1Ft5 · 2025-06-04

**Clarity:** 4
**Significance:** 3
**Originality:** 3
**Rating:** 5
**Confidence:** 4

**Summary:**

This paper introduces Screen-Aligned Primitives (SAP), a new rendering framework that solves the depth-sorting problem in 3D Gaussian Splatting. Instead of using 3D Gaussians that often overlap and require approximate sorting, SAP predicts view-dependent 2D Gaussians that are always aligned with the image plane. This allows exact per-pixel depth sorting.
The primitives are predicted from a set of 3D anchor Gaussians using a lightweight decoder network, which is conditioned on the viewing direction and some input features.
Experiments show that SAP achieves state-of-the-art rendering quality with real-time performance, eliminating artifacts like popping, while maintaining a consistent and compact 3D representation.

**Questions:**

1. The method does not model surface normals or enforce geometric priors, making it difficult to extract meshes. Do the authors have an idea on how to integrate smooth surface constraints to enable mesh extraction with SAP?

2. The tables include publication year and venue, which are not particularly informative for evaluating method performance. I recommend replacing them with the memory consumption, training time, and inference time, as done in the original 3DGS paper.
Including such metrics would better support the real-time rendering claims and improve the empirical comparison.

**Ethical Concerns:**

["NO or VERY MINOR ethics concerns only"]

**Final Justification:**

After reviewing the rebuttal, I find that the authors have fully addressed all of my original concerns. They provided a clear plan for integrating smooth surface constraints to enable mesh extraction, demonstrating awareness of potential geometric limitations and outlining concrete future directions. They also agreed to revise the tables to replace publication year/venue with more relevant performance metrics (e.g., memory usage, FPS, storage size).

The proposed Screen-Aligned Primitives framework remains technically solid, addresses a well-known limitation of 3D Gaussian Splatting, and achieves high visual quality with real-time performance. The authors’ clarifications further strengthen the significance and reproducibility of the work. I maintain my original positive assessment and accept recommendation.

**Limitations:**

yes

**Paper Formatting Concerns:**

/

**Quality:**

3

**Strengths And Weaknesses:**

+) SAP addresses a well-known issue in Gaussian Splatting: the incorrect ordering of overlapping Gaussians, which leads to popping artifacts due to sorting based on the center position rather than per-pixel depth. In contrast to concurrent methods like EVER, which perform per-pixel sorting at the cost of speed, SAP proposes a neural network that predicts view-dependent 2D Gaussians aligned with the camera's image plane. As a result, a simple global sort by the primitive centers yields correct per-pixel ordering. This approach is effective, and it leads to improved visual quality over prior work.
The figures are clear and well-designed, making the core ideas and architecture easy to understand.
Overall, this is a good submission that improves upon existing methods while maintaining conceptual clarity.

-) SAP introduces view-dependent primitives, meaning the set of rendered splats (and their properties) change with the camera angle. This is handled by the learned 3D-consistent decoder, but it also adds complexity and potential points of failure. If the decoder does not perfectly maintain consistency, there could be flickering or popping of primitives when the viewpoint changes.

-) By relying on a neural network to predict Gaussians conditioned on the camera direction, the method moves away from an explicit 3D representation populated with fixed Gaussians. Instead of maintaining a globally defined set of 3D primitives, the scene is now implicitly represented through a view-conditioned function, making it harder to reason about or extract a concrete 3D structure. In addition, the improvements over other baselines remain relatively marginal in terms of standard quantitative metrics such as PSNR, SSIM, and LPIPS.

-) The approach does not incorporate surface normals or an explicit surface prior, focusing only on rendering quality rather than geometric fidelity. The authors acknowledge that, in the absence of normal constraints, extracting a mesh from the representation is difficult. While mesh extraction is beyond the scope of novel view synthesis, this limitation reduces the applicability of SAP in scenarios that require explicit 3D geometry (though this is a minor concern given the paper’s emphasis on image-based rendering). It would be interesting for future work to explore ways of integrating smooth surface constraints, such as learned normals,... , to enable consistent and accurate mesh extraction from the representation. Are there any ideas on how future work could incorporate smooth surface constraints to enable mesh extraction?

-) In my opinion, the year and conference are not important or necessary to include in the table. I would recommend removing them and instead adding metrics such as memory usage, training time, and inference time (as presented in the 3DGS paper), as these offer significantly more meaningful insights into the performance of the methods.

-) I would also add Zip-NeRF as one of the baselines, as it is currently considered the state of the art on the Mip-NeRF360 dataset.

-) Some minor comments:

L248: Typo — "Implenmentation Details" should be corrected to "Implementation Details."

L129: I would avoid starting the section with “To overcome this limitation,” as the specific limitation being referred to is not clearly introduced in this section, and the reader may not understand what it refers to.

This is just a personal preference, but I would suggest placing the qualitative results of SAP next to the ground truth. This would make visual comparisons easier for the reader.

---

> ### Author Rebuttal · Authors · 2025-07-29
>
> We thank the reviewer for the valuable comments. Below is our supplementary response to the reviewer’s concerns.
>
> # Q1. How to integrate smooth surface constraints to enable mesh extraction with SAP?
> We fully recognize the value of integrating smooth surface constraints to support mesh extraction, which stands as a key direction for our future work. We are also more than willing to discuss ideas and prospects in this regard with the reviewers. Leveraging the characteristics of the SAP method, potential specific approaches are as follows:
>
> **1. Introducing Learnable Geometric Metrics into the Planar Primitive Framework**
>
> Given that SAP’s core lies in its "explicit anchors + neural network-driven primitive representation," we can introduce an **additional module for learning geometric attributes for each planar primitive.** For instance, drawing inspiration from the approach in normal-gs and spec-gs—where neural networks directly learn surface normals—we can specifically predict geometrically meaningful metrics such as **normals, signed distance functions (SDF), or depth.** These metrics inherently encode spatial continuity constraints (e.g., normal consistency between adjacent primitives), which can indirectly enforce "smooth surface constraints."
>
> **2. Ensuring primitive correlation and geometric consistency**
>
> Unlike purely implicit representations, the planar primitives in SAP exhibit local spatial correlation, as defined by the positional relationships of anchors. When learning geometric metrics, we will **strengthen constraints between adjacent primitives (e.g., normal gradient penalties, SDF continuity losses)** to ensure that the overall geometry formed by combining primitives is smooth and consistent. This provides a critical foundation for subsequent mesh extraction (e.g., via the Marching Cubes algorithm applied to the SDF field).
>
> **3. Compatibility with the advantages of neural rendering**
>
> Notably, such an extension will not compromise SAP’s strengths in rendering quality: **planar primitives will still serve as carriers of appearance textures, while the newly added geometric metrics will only be used to constrain structural smoothness.** This resembles the division of labor in NeRF, where "radiance fields encode appearance and density fields implicitly represent geometry." **However, the explicit anchor in SAP enable more direct control over geometric accuracy, facilitating targeted optimization for the consistency required for mesh extraction.**
>
> **4. Future Directions: Bridging SAP and Feed-forward 3D Models**
>
> While this may be slightly beyond the scope of the current work, we appreciate the opportunity to share some of our future research directions in 3D representations. Recent advances in feed-forward models, such as VGGT and LSVM, have inspired us to reflect on the nature of our SAP framework. In essence, SAP can be viewed as a per-scene optimization variant of LSVM, where multi-view observations are directly used to represent 3D without relying on strong geometric priors.
>
> However, due to the lack of data priors in per-scene optimization, geometric anchors (e.g., those introduced in our work) remain necessary to stabilize and guide the learning process. Looking forward, we envision two promising directions:
> (1) Incorporating large-scale pretrained models (e.g., depth or normal predictors) to provide strong priors that improve the reconstruction fidelity and geometric consistency;
> (2) Towards feed-forward Gaussian reconstruction, where existing methods typically generate per-pixel 3D Gaussians. We believe that combining the principles of LSVM and SAP could enable a novel 3D representation paradigm that transcends purely pixel-wise modeling and incorporates more structured geometric reasoning.
>
> # Q2. Optimization of table content
>
> Yes, we acknowledge this as a minor oversight on our part. The publication year and venue do not contribute significantly to the core information conveyed in the table, and therefore have limited impact on its clarity or usefulness. In the final published version, we plan to replace these entries with more informative metrics such as **inference FPS** (to highlight our advantage over NeRF-based methods) and **storage size or point count** (to emphasize our inheritance from Scaffold-GS and demonstrate our storage efficiency compared to 3DGS). We have already reported some of these performance results in Table 1 of the appendix, and we will include additional metrics to provide a more comprehensive evaluation.We provide a comparison of computational resources between our method and the main baselines. All experiments were conducted on a single NVIDIA L40S GPU.
>
> | Tanks&Temples | PSNR   | SSIM   | LPIPS  | Mem(M)  | fps    |
> |---------------|--------|--------|--------|---------|--------|
> | 3DGS          | 23.89  | 0.851  | 0.17   | 407.1M  | **103.5**  |
> | Scaffold-GS   | 24.45  | 0.864  | 0.155  | 169.5M  | 90.1   |
> | SAP(ours)     | **24.98**  | **0.868**  | **0.151**  | **165.2M**  | 88.2   |
>
> We agree with the reviewer’s suggestion to include Zip-NeRF as a baseline. Zip-NeRF is indeed the current state-of-the-art on the Mip-NeRF 360 dataset and should be considered for fair comparison. At the same time, we would like to highlight that **one of the main advantages of the 3DGS-based methods over Zip-NeRF is the significantly faster rendering speed**. Here, we provide a brief comparison between our method and Zip-NeRF, and **we commit to including Zip-NeRF as a baseline in the final version of the paper**.
>
> | Methods          | PSNR   | SSIM   | LPIPS  | Mem     | fps |
> |---------------|--------|--------|--------|---------|----------|
> | zip-nerf      | **28.54**  | 0.829  | 0.218  | 26.82G  | <3 |
> | 3DGS-MCMC     | 28.00     | 0.831  | **0.188**  | 28.86G  | ---|
> | SAP(ours)     | 28.05  | **0.835**  | 0.208  | **11.25G**   | **83.48** |
>
> It is important to note that due to differences in downsampling strategies, **the data reported in the Zip-NeRF paper are not directly comparable to those commonly used in 3DGS methods.** Specifically, if the downsampling factor employed by Zip-NeRF—namely, 4× downsampling for outdoor scenes—is applied, the performance metrics of the 3DGS-base method would actually exceed the numbers reported in our paper.
>
> |              | indoor(PSNR) | outdoor(PSNR) |
> |--------------|--------------|---------------|
> | zip-nerf     | 32.25        | **25.56**         |
> | SAP(ours)    | **32.28**        | 24.63         |
>
> We also acknowledge that our method, along with many existing 3DGS-based approaches, does not perform as well as Zip-NeRF on outdoor scenes in the Mip-NeRF 360 dataset.
>
> # Q3. Minor comments:
> 1. Thank you for pointing out the typo at L248. We will correct "Implenmentation Details" to "Implementation Details" in the revised manuscript. \
> 2. Regarding the wording at L129, we agree that starting the section with “To overcome this limitation” without a clear prior mention may confuse readers. We will revise this sentence to explicitly state that **the limitation refers to the popping artifacts of Gaussian primitives and inaccuracies in their sorting order, thereby improving clarity.** \
> 3. We appreciate the suggestion to place the qualitative SAP results alongside the ground truth for easier visual comparison. We will reorganize the figure layouts accordingly in the final version to improve readability.
>
> >  [1] Jonathan T Barron, Ben Mildenhall, Dor Verbin, Pratul P Srinivasan, and Peter Hedman. Zip-NeRF: Anti-Aliased Grid-Based Neural Radiance Fields. ICCV, 2023. \
> >  [2] Kerbl, B., Kopanas, G., Leimkühler, T., Drettakis, G.: 3d gaussian splatting for real-time radiance field rendering. ACM Transactions on Graphics 42(4) (2023) \
> > [3] Tao Lu, Mulin Yu, Linning Xu, Yuanbo Xiangli, Limin Wang, Dahua Lin, and Bo Dai. Scaffold-gs: Structured 3d gaussians for view-adaptive rendering. \
> > [4] Haian Jin, Hanwen Jiang, Hao Tan, Kai Zhang, Sai Bi, Tianyuan Zhang, Fujun Luan, Noah Snavely, and Zexiang Xu. Lvsm: A large view synthesis model with minimal 3d inductive bias. arXiv preprint arXiv:2410.17242, 2024. \
> > [5] S. Kheradmand, D. Rebain, G. Sharma, W. Sun, Y.-C. Tseng, H. Isack, A. Kar, A. Tagliasacchi, and K.M.Yi, “3d gaussian splatting as markov chain monte carlo,”Advances in Neural  Information Processing Systems, vol.

---

### Official Review · Reviewer_WHxg · 2025-06-10

**Clarity:** 3
**Significance:** 3
**Originality:** 3
**Rating:** 4
**Confidence:** 3

**Summary:**

Overall, this submission focuses on addressing the popping artifects in 3D-GS. It proposes to deal with this via following Scaffold-GS's anchor-based pipeline and further guiding the generation of screen-aligned primitives.

**Questions:**

Overall, the reviewer currently votes for borderline accept while he still has the following concerns.

1. The authors propose to use parallel thin slice instead of 3D Gaussians as their primitives and show some appealing properties of this choice. Despite this, intuitively, parallel thin slices can still be regarded as primitives that has less representation freedom than 3D Gaussians. The reviewer is interested to see more discussion from this perspective.

2. For example, one potential disadvantage of parallel thin slices can be that their multi-view inconsistency. While the authors claim here and there that they use a "3D-consistent" decoder. The reviewer is still a bit confused about how 3D (multi-view) consistency can be guaranteed for parallel thing slices. The authors are appreciated to theoretically (or at least emphrically) explain this part.

3. In the related works, the authors mention that those methods in the early stage ([44, 45]) do not suffer from popping artifacts. Considering this, the authors are appreciated to better discuss, why such ideas cannot be directly incorporated into the recent 3DGS framework to handle popping artifacts?

4. In line 129, the "this" in "to overcome this limitation" seems to point to nowhere.

5. in Section 4.1, the authors mention that they decide to use SH for encoding. Yet, one typical problem in SH can be there parameter demanding property (e.g., when used to represent color for Gaussian kernels in 3D-GS). In this case, the reviewer is wondering if the usage of SH in this method's context also suffer it from high memory overhead?

6. In line 167, the authors mention that the activation is "such as ReLU". Such ambiguity is not good for reproduction and thus authors are thus suggested to make this part more clear.

In summary, the reviewer currently votes for borderline accept for this submission while still holding the above concerns.

**Ethical Concerns:**

["NO or VERY MINOR ethics concerns only"]

**Final Justification:**

After reading the rebuttal, I believe that my concerns are largely solved. I thus keep my positive rating. Especially, I believe that the authors explain their intuition more clearly in their response to Q1 and I highly suggest the camera-ready to be edited based on this answer to better show the merit of the paper.

**Limitations:**

Yes

**Paper Formatting Concerns:**

N.A.

**Quality:**

3

**Strengths And Weaknesses:**

Strength:

This submission is easy to follow. Meanwhile, the reviewer also feels that the methods proposed hold their merits.

Weaknesses:

(See the questions section for more details).

---

> ### Author Rebuttal · Authors · 2025-07-30
>
> We thank the reviewer for the valuable comments. Below is our supplementary response to the reviewer’s concerns.
>
> # Q1. Discussion on the Representational Capacity of SAP Compared to 3D Gaussians
>
> This is an excellent question and, in fact, one of the core insights that directly motivated our initial idea. Our baseline, Scaffold-GS, is a neural network-driven approach that generates a set of 3D Gaussians for each individual viewpoint. In other words, the 3D primitives are generated dynamically per view, and are different for each camera pose.
>
> This led us to a key observation: **if we are generating entirely different primitives for every view, and each viewpoint only observes one face of the 3D Gaussian, then why do we need a full 3D representation in the first place?** This insight inspired us to explore using 2D primitives instead.
>
> Compared to surface-aligned 2D Gaussians, which still possess 3D degrees of freedom and can **suffer from overlapping and blending artifacts**, our view-aligned planar primitives offer a more constrained yet more targeted representation. Since **the rendering viewpoint is fixed per frame, this effectively fixes one axis of rotation.** Therefore, using planar primitives aligned with the view direction **not only simplifies depth sorting but also makes more efficient use of the viewpoint direction as an input to the neural network.**
>
> # Q2. Clarification on How 3D (Multi-View) Consistency Is Ensured with SAP
>
> From an architectural perspective, our method is composed of a 3D decoder and a SAP (Screen-Aligned Primitive) renderer. The **3D (multi-view) consistency is primarily ensured by the 3D decoder,** which encodes scene content in a consistent volumetric space.
> As illustrated in Appendix Figure 6, **we visualize the decoded attributes of SAPs across multiple views.** It can be seen that, thanks to the 3D decoder, the attributes of the generated primitives exhibit strong continuity across different viewpoints. In fact, **for a given SAP ID, its projections in different views demonstrate high consistency in both spatial location and appearance, even though their orientations vary with the view.**
>
> This can also be interpreted from another perspective: **the underlying neural field implicitly defines a complex, view-agnostic primitive function in 3D space. Each viewpoint observes only a “slice” of this function, which we approximate as a 2D Gaussian. We do not need to explicitly reconstruct the full 3D geometry of this function — instead, we extract only the per-view 2D shape necessary for rendering.**
>
> In this design, **the SAP primitives primarily serve as carriers of texture and appearance**, effectively bridging the gap between implicit neural rendering and explicit, structured primitive-based rendering.
>
> At the microscopic level, the primitives we construct are all aligned with the screen. However, in practice, our method is an unstructured **primitive-based representation that conceptually resembles a hybrid between neural rendering and explicit anchor-based primitives**, rather than relying solely on explicit geometry to fit object surfaces. Instead of modeling object surfaces with complex geometric primitives, **our approach represents appearance through the collective blending of a large number of simple, planar primitives during rendering.** For example, a complex curved surface is not modeled by a single curved primitive, but rather emerges from the layered and aggregated contributions of thousands of flat primitives. As a result, the overall representational capacity of the model far exceeds the limitations of any single primitive's shape.
>
> In addition, Figure 3 in our main paper demonstrates reconstructions on a variety of challenging scenes, and **our supplementary material further provides 3D visualizations along continuous camera trajectories.** These results effectively validate the 3D consistency of our approach in practice. We additionally include experiments on the NeRF-Synthetic dataset, which **contains several scenes with strong non-Lambertian effects**. In particular, the Mic, Materials, and Drums scenes exhibit complex view-dependent reflections and specularities. The supplementary experimental results are presented below.
>
> | Scene     | PSNR  | SSIM  | LPIPS  |
> |-----------|-------|-------|--------|
> | mic       | 36.75 | 0.991 | 0.009  |
> | materials | 30.37 | 0.954 | 0.051  |
> | drums     | 26.31 | 0.945 | 0.049  |
>
>
> # Q3. Why can't traditional methods be applied to mitigate popping artifacts in 3D Gaussian Splatting (3DGS)?
>
> There exists a **fundamental difference between Voltx[44] and 3DGS** in terms of their mechanisms and suitability for handling popping artifacts, which stems from the essential differences in their **data representations and rendering logic**:
>
> **Voltx** is designed for continuous, regularly structured volumetric data. It partitions space using **uniform thin slices** and leverages **3D texture hardware interpolation** to enable smooth sampling. Colors and opacities are composited layer by layer in a **back-to-front** manner. The key to mitigating popping artifacts in Voltx lies in its **“uniform discretization of continuous fields”** — with **fixed inter-slice distances** and **interpolable information**, the sampled results across neighboring frames **transition smoothly**. This approach is well-suited for rendering **structured volumetric data** such as **CT and MRI scans**.
>
> In contrast, **3DGS** represents the scene as a large set of **independent 3D Gaussian primitives**, and the final rendering is a **2D accumulation of their projections**. Popping artifacts in 3DGS are primarily caused by **visibility discontinuities** (i.e., sudden appearance/disappearance of Gaussians) or **attribute discontinuities** (i.e., abrupt changes in projected size or shape). Since Gaussians are **discrete primitives**, they cannot benefit from **“partial inclusion in a slice”** to achieve smooth transitions, as voxels do.
>
> Therefore, **Voltx’s continuous layer-based strategy cannot be directly applied to the discrete nature of 3DGS**. While Voltx addresses **discontinuities arising from coarse sampling** in continuous fields, 3DGS requires solutions tailored to **discontinuities in discrete dynamic elements**. These are **fundamentally different problems** requiring **distinct approaches**.
>
> # Q4. What specific limitation does "this" refer to in line 129? The antecedent is unclear.
>
> We agree that starting the section with “To overcome this limitation” without a clear prior mention may confuse readers. We will revise this sentence to explicitly state that **the limitation refers to the popping artifacts of Gaussian primitives and inaccuracies in their sorting order, thereby improving clarity.**
>
> # Q5. Does the use of spherical harmonics (SH) in our method introduce high memory overhead?
>
> It is important to clarify that we employ **hard-coded spherical harmonics (SH)**, meaning that the directional information is lifted via **mathematical formulas**, introducing only **a few dozen additional variables**, which is **negligible in terms of memory overhead**. Meanwhile, when using SH encoding, we **adjust the input dimensionality** of the neural network, which results in a **slight change in memory usage**. However, in practice, the **majority of memory consumption** comes from storing the **anchor point cloud**. Below, we provide a **comparison of the memory usage** between the MLP and the point cloud components across different methods.
>
> | Method | Anchor Num (k) | Anchor Mem (MB) | Network (MB) |
> |--------|----------------|------------------|---------------|
> | No SH  | 114.65         | 385.3            | 0.05          |
> | SH     | 114.33         | 384.2            | 0.05          |
>
> From the table, it can be seen that the **storage mainly depends on the number of anchors**, and is **almost unaffected by whether spherical harmonics encoding is used**.
>
> # Q6. Clarification on Activation Function (Line 167)
>
> Thank you for pointing this out. We acknowledge that the phrase "such as" here may cause misunderstanding. In fact, we use **ReLU** as the activation function for all intermediate layers in the network. More detailed network architecture is provided in **Appendix B.3** for readers to reproduce our work. We commit to revising this in the final version of the paper and releasing the source code to facilitate further reproducibility.
>
> > Tao Lu, Mulin Yu, Linning Xu, Yuanbo Xiangli, Limin Wang, Dahua Lin, and Bo Dai. Scaffold-gs: Structured 3d gaussians for view-adaptive rendering. \
> > A. Van Gelder and K. Kim,  “Direct volume rendering with shading via three-dimensional textures. \
> > Binbin Huang, Zehao Yu, Anpei Chen, Andreas Geiger, and Shenghua Gao. 2d gaussian splatting for geometrically accurate radiance fields.

---

> > ### Comment · Reviewer_WHxg · 2025-08-07
> >
> > Hi authors,
> >
> > I am sorry that previously, I thought you can see the final justification and I thus left my feedback to your rebuttal there.
> >
> > As mentioned there, after reading the rebuttal, I believe that my concerns are largely solved. I thus keep my positive rating. Especially, I believe that the authors explain their intuition more clearly in their response to Q1 and I highly suggest the camera-ready to be edited based on this answer to better show the merit of the paper.

---

> > > ### Author Response · Authors · 2025-08-07
> > >
> > > We are delighted to hear that our response has addressed your concerns. Thank you also for the clarification regarding your previous feedback. We especially value your suggestion regarding our answer to Q1. We agree that it helps clarify the core intuition of our paper, and we will be sure to revise the camera-ready version to incorporate this explanation, as you highly recommended.
> > > Thank you once again for your constructive guidance in helping us improve our paper.

---

### Official Review · Reviewer_F3jw · 2025-06-24

**Clarity:** 3
**Significance:** 3
**Originality:** 2
**Rating:** 4
**Confidence:** 4

**Summary:**

The paper introduces Screen-Aligned Primitives (SAP) to improve 3D Gaussian Splatting rendering. The primary issue with 3DGS is "popping artifacts" caused by an inaccurate sorting of primitives that doesn't account for their thickness and overlapping regions. To solve this, the authors propose a method that generates primitives constrained to be parallel to the image plane for any given viewpoint. Because all primitives are parallel, a single global depth sort becomes sufficient to achieve a correct per-pixel rendering order, fundamentally eliminating the sorting problem without extra computational cost. The framework uses a neural decoder to generate these view-specific 2D primitives conditioned on anchor features and viewing direction, which allows for an error-free projection and the use of more flexible rendering kernels.

**Questions:**

- The ablation study in Table 2 shows a curious inconsistency where Screen-Aligned Primitives improve PSNR over the baseline, yet score worse on SSIM and LPIPS in most intermediate configurations, which warrants a deeper analysis of the qualitative trade-offs between the two representations.

- To better understand the method's generalization capabilities, a robustness study would be valuable, such as evaluating performance with sparser input views, especially given the framework's similarity to multi-plane image rendering which can be sensitive to training data density.

**Ethical Concerns:**

["NO or VERY MINOR ethics concerns only"]

**Final Justification:**

Authors rebuttal addressed most of my concerns. Given the experiment results I do think this is an insightful improvement of 3DGS/2DGS. I'm raising my score to BA.

**Limitations:**

See questions and weakness.

**Quality:**

3

**Strengths And Weaknesses:**

Strengths:
+ The paper is well-motivated and clearly written, making its contributions easy to understand.
+ The core idea of using Screen-Aligned Primitives is straightforward and intuitive.
+ This approach is conceptually similar to multi-plane image (MPI) based rendering, providing a strong rationale for why it effectively eliminates overlap artifacts and allows for a simple, global sorting pass.

Weakness:
The method's strong similarity to multi-plane image (MPI) based rendering suggests a potential limitation, common to such approaches, which is its dependency on the training data's density and distribution. The framework's performance is likely to be strongest when rendering novel views that are interpolated between the dense viewpoints seen during training. However, in the qualitative results, authors did not illustrate the distribution of training and testing views, the video demos in supp are also rendered from simply trajectories. Similarly, the approach may struggle if the training data itself is sparse, as the multiplane-based representation might not generalize well across large, unseen gaps between viewpoints. The paper does not fully investigate these challenging scenarios.

---

> ### Author Rebuttal · Authors · 2025-07-29
>
> We thank the reviewer for the valuable comments.
>
> # Q1. Comparison with Multi-Plane Image (MPI)-based Rendering Models and Robustness under Sparse Viewpoints.
> Regarding the reviewer's concern about the quality of view synthesis under sparse viewpoints, we clarify this issue from both theoretical and experimental perspectives as follows:
> 1. **Theoretical Analysis:**
>
> The MPI method is less capable of supporting observations from the side or back relative to the reference views. In contrast, SAP does not rely on selected reference views but instead **models the entire 3D space in an equitable manner, thereby enabling observations from test viewpoints far from the training viewpoints. Meanwhile, SAP uses explicit anchors to represent the scene, which endows it with relatively clear geometric properties. The planes here serve more as appearance textures.**
>
> Furthermore, although both MPI and SAP represent 3D space with primitives parallel to the camera image plane during modeling, they differ in key aspects. **MPI is a dense volumetric representation, incurring high memory overhead, whereas SAP is an adaptively optimized sparse representation with lower memory usage.** Additionally, **the number of depth layers in MPI is a manually set hyperparameter**, lacking sufficient flexibility to fit the true continuous 3D space. In contrast, the **adaptive splitting and pruning mechanism** of SAP allows it to learn an appropriate complexity and continuously varying attributes, thereby fully representing the target scene.
>
> 2. **Experimental Validation:**
>
> Addressing the reviewer's concern regarding robustness under sparse viewpoints, we recognize this as a highly valuable research area, with numerous 3DGS-based works already making improvements in this regard. For 3DGS-based methods, we believe that dense viewpoints significantly benefit the reconstruction process. Additionally, we supplement here with experiments to demonstrate that **our primitive framework does not compromise performance in reconstructing sparse viewpoints (compared to previous methods), even without specific optimization for sparse viewpoints**.
>
> All the following experiments were conducted using a single L40S GPU. We performed experiments on the Tanks & Temples dataset. Specifically, we followed **the original training and test set splits, then used 12.5%, 25%, 50%, and 100% of the training viewpoints for reconstruction, respectively, and tested using all test viewpoints**. The experimental results are shown in the table below. Our experiments indicate that our method outperforms our baseline even when the number of viewpoints is significantly reduced.
>
> | Training view ratio   | Method           | psnr   | ssim   | lpips  |
> |--------------|---------------|--------|--------|--------|
> | 12.50%       | 3dgs          | 13.57  | **0.479**  | **0.485**  |
> |              | scaffold-gs   | 13.62  | 0.418  | 0.486  |
> |              | **SAP(ours)**   | **13.84**  | 0.448  | 0.508  |
> | 25%          | 3dgs          | 16.05  | **0.574**  | **0.397**  |
> |              | scaffold-gs   | 16.22  | 0.553  | 0.422  |
> |              | **SAP(ours)**   | **16.44**  | 0.568  | 0.419  |
> | 50%          | 3dgs          | 22.04  | 0.803  | 0.199  |
> |              | scaffold-gs   | 22.46  | 0.808  | **0.193**  |
> |              | **SAP(ours)**   | **22.61**  | **0.809**  | 0.203  |
> | 100%         | 3dgs          | 23.82  | 0.853  | 0.169  |
> |              | scaffold-gs   | 24.14  | 0.859  | 0.163  |
> |              | **SAP(ours)**   | **24.97**  | **0.867**  | **0.151**  |
>
> Additionally, regarding the reviewer's comment on the similarity between our method and MPI—specifically, that favorable results are often obtained only when testing viewpoints lie between training viewpoints—we acknowledge this as a critical research issue and a well-known pain point in current reconstruction methods. To address this, we conducted supplementary experiments.
>
> As the reviewer anticipated, **the performance is significantly better when test viewpoints fall within the interpolation range of training viewpoints compared to the extrapolation range. We further found that this limitation is common to all 3DGS-based methods. Even so, our method still outperforms previous approaches under such conditions.**
>
> The experimental results are presented below. We conducted experiments on the train scenes of the Tanks & Temples dataset, where the numbers in parentheses following the percentages indicate the number of training viewpoints. **"Interpolation"** refers to test viewpoints lying within the interpolation range of training viewpoints, and **"Extrapolation"** refers to those in the extrapolation range. Our division criterion is: a test viewpoint is categorized as "Interpolation" if it lies within the convex hull formed by any two training viewpoints.
>
> | Training view ratio   | Method        | Interpolation |        | Extrapolation |        | ALL |        |
> |:----------------------:|:-------------:|:--------------:|:------:|:--------------:|:------:|:-----:|:------:|
> |                |             | views                 | psnr   | views                 | psnr   | views       | psnr   |
> | 12.5%（32）    | **SAP(ours)**      | 12                    | **19.15**  | 25                    | 9.71   | 37          | **12.77**  |
> |                | Scaffold-GS | 12                    | 18.92  | 25                    | 9.16   | 37          | 12.33  |
> |                | 3DGS | 12                    | 17.93  | 25                    | **9.81**   | 37          | 12.44  |
> | 25%（65）      | **SAP(ours)**       | 16                    | **22.51**  | 21                    | **10.01**  | 37          | **15.42**  |
> |                | Scaffold-GS | 16                    | 22.18  | 21                    | 9.99   | 37          | 15.26  |
> |                | 3DGS | 16                    | 21.51  | 21                    | 9.95   | 37          | 14.95 |
> | 50%（131）     |**SAP(ours)**      | 37                    | **20.58**  | 0                     | 0      | 37          | **20.58**  |
> |                | Scaffold-GS | 37                    | 20.50  | 0                     | 0      | 37          | 20.50  |
> |                | 3DGS | 37                    | 20.01  | 0                     | 0      | 37          | 20.01  |
> | 100%（263）    | **SAP(ours)**       | 37                    | **23.66**  | 0                     | 0      | 37          | **23.66**  |
> |                | Scaffold-GS | 37                    | 22.48  | 0                     | 0      | 37          | 22.48  |
> |                | 3DGS | 37                    | 22.02  | 0                     | 0      | 37          | 22.02  |
>
>
> # Q2.Clarification on SAP's performance in terms of PSNR, SSIM, and LPIPS metrics.
>
> Theoretically, our method is a **pixel-level hybrid approach**, meaning that rendering is achieved by stacking and blending numerous small local planes. This inherently makes it well-suited for pixel-wise comparison metrics like PSNR, and our method is also optimized based on **MSE loss**. Our research has found that in most cases, PSNR and SSIM are positively correlated. However, since our method is a pixel-blending approach, it does not maintain a static point cloud. Instead, it dynamically generates planar primitives according to the rendering viewpoint and then performs pixel-level alpha-blending, which is naturally suitable for the PSNR metric.
>
> At the same time, because our method uses **dynamic view-dependent primitives**, **edge and other structural information in the scene may be damaged under some special viewpoints, resulting in lower SSIM.** In conclusion, for our primitive-blending model, which is not a specific 3D structure, a small three-dimensional object in space may be formed by the blending of a large number of primitives. **This method, when the primitive density is high enough and the distribution is reasonable — that is, our final scheme (with maximum densification) — will achieve the best performance.** When the density is insufficient, they cannot blend to form sufficiently detailed structural information.
>
> We have also observed that for metrics such as LPIPS, improvements in densification often lead to better performance. This finding is consistent with the conclusions drawn in Pixel-GS. We speculate that more primitives can enhance the overall visual perception; however, they may simultaneously introduce subtle pixel-wise differences. A close look at **the last image in the first row of Figure 3** reveals that when zoomed in, there are indeed some pixel-level errors, but it remains the best in terms of overall human visual perception. Our ablation study in Table 2 also shows that **the maximum densification method can effectively improve the LPIPS metric across all approaches**. We have supplemented additional experiments on the train scenes of the Tanks & Temples dataset here.
>
> | psnr   | ssim   | lpips  | SAP num(k) |
> |--------|--------|--------|------------|
> | **23.54**  | **0.841**  | **0.168**  | **1693**       |
> | **23.56**  | **0.840**   | **0.162**  | **1020**       |
> | 23.52  | 0.838  | 0.178  | 789        |
> | 23.31  | 0.828  | 0.188  | 679        |
> | 23.12  | 0.823  | 0.207  | 588        |
>
> The experiments here demonstrate that when the densification process is improved—most simply by **increasing the number of primitives—the LPIPS metric shows a more significant improvement.** This is consistent with the results of our ablation study in Table 2.
>
> >  [1] Z.Zhang, W.Hu, Y.Lao, T.He, and H.Zhao, “Pixel-gs: Density control with pixel-aware gradient for 3d gaussian splatting,” in European Conferenceon Computer Vision.  \
> >  [2] Kerbl, B., Kopanas, G., Leimkühler, T., Drettakis, G.: 3d gaussian splatting for real-time radiance field rendering. ACM Transactions on Graphics 42(4) (2023) \
> > [3] Tao Lu, Mulin Yu, Linning Xu, Yuanbo Xiangli, Limin Wang, Dahua Lin, and Bo Dai. Scaffold-gs: Structured 3d gaussians for view-adaptive rendering.

---

> > ### Comment · Reviewer_F3jw · 2025-08-06
> >
> > Thank authors for their thorough rebuttal. Most of my concerns were resolved given the additional experiments..

---

### Official Review · Reviewer_zbg4 · 2025-06-30

**Clarity:** 3
**Significance:** 2
**Originality:** 3
**Rating:** 4
**Confidence:** 4

**Summary:**

This paper addresses the problem of depth-sorting inaccuracies in 3D Gaussian Splatting. Depth-sorting inaccuracies cause popping artifacts and rendering inconsistencies due to overlapping anisotropic Gaussians. The paper proposes Screen-Aligned Primitives (SAP) as a solution. SAP generates per-view, plane-parallel primitives to eliminate per-pixel overlap errors and enables exact sorting without additional computational cost. The design is supported by a 3D-consistent decoder and flexible kernel choices. It is validated through comprehensive experiments, which show improved rendering quality and real-time performance.

**Questions:**

1. How does the proposed screen-aligned primitive design handle highly complex geometry or scenes with strong non-Lambertian appearance, where local shape details or view-dependent reflections may challenge the planarity and directional consistency assumptions? More experiments and discussion are needed to clarify how the method performs for materials with transparent or semi-transparent properties, which may not be well approximated by a single opaque plane.
2. Could the authors discuss how the method performs in scenes with large depth ranges or multiple layers of fine occlusion, where a single global sorting order may not fully resolve local ordering conflicts?
3. Clarify how the maximum-gradient densification strategy deals with noisy regions or high-frequency texture boundaries, and whether it risks adding unnecessary primitives that increases computation without clear quality benefits.
4. Please provide more information on training and inference memory usage, runtime cost, and possible trade-offs between sorting precision.

**Ethical Concerns:**

["NO or VERY MINOR ethics concerns only"]

**Final Justification:**

After reading the authors' rebuttal and the comments from other reviewers, I am basically positive about the paper.  Thus my assessment maintains the original rating: BA.

**Limitations:**

The paper discusses the main limitations and outlines possible directions for future improvement.

**Paper Formatting Concerns:**

NIL

**Quality:**

3

**Strengths And Weaknesses:**

Strengths:

-- The proposed approach is intuitive and easy to understand. Introducing screen-aligned primitives allows per-pixel ordering with low computational cost, which well addresses the depth-sorting issue in 3D Gaussian Splatting.

-- The method is compatible with existing frameworks and supports flexible kernel designs without projection bias.

-- Experimental results on several benchmarks show that the method improves rendering quality and consistency compared to related baselines.


Weaknesses:

-- The method assumes that all primitives can be well approximated as screen-parallel slices. This may not be always true, limiting its adaptability to highly complex or non-Lambertian scenes.

-- The current design focuses mainly on radial basis kernels and does not yet explore asymmetric or more diverse kernel shapes. Some implementation details, such as potential trade-offs between sorting precision and densification cost, could be elaborated further.

---

> ### Author Rebuttal · Authors · 2025-07-30
>
> We thank the reviewer for the valuable comments. Below is our supplementary response to the reviewer’s concerns.
>
> # Q1. How does SAP handle highly complex geometries or scenes with strong non-Lambertian appearances, where local shape details or view-dependent reflections may challenge the assumptions of planarity and directional consistency?
>
> This is indeed a representative and potentially misleading concern for readers. We address your concern regarding the “screen-aligned slicing” assumption from both experimental and theoretical perspectives:
>
> **1. Theoretical Analysis:**
>
> At the microscopic level, the primitives we construct are all aligned with the screen. However, in practice, our method is an unstructured **primitive-based representation that conceptually resembles a hybrid between neural rendering and explicit anchor-based primitives**, rather than relying solely on explicit geometry to fit object surfaces. Instead of modeling object surfaces with complex geometric primitives, **our approach represents appearance through the collective blending of a large number of simple, planar primitives during rendering.** For example, a complex curved surface is not modeled by a single curved primitive, but rather emerges from the layered and aggregated contributions of thousands of flat primitives. As a result, the overall representational capacity of the model far exceeds the limitations of any single primitive's shape.
>
> This is analogous to how 2D images are composed: although each pixel is a simple square, a high-resolution image can express complex structures and details through dense sampling. Similarly, our primitives serve as the minimal representational units in 3D space. In regions with intricate geometry, our SAP and adaptive density control module automatically allocate a large number of primitives to ensure sufficient coverage and accurate appearance modeling. As illustrated in **the point distribution visualization in Appendix Table 4**, dense clusters of small primitives are concentrated in geometrically complex regions.
>
> **2. Experimental Validation:**
>
> The datasets we selected are **among the most commonly used and challenging benchmarks in this field.** The qualitative results we present already cover the two types of scenarios mentioned by the reviewer. For instance, **the first row of Figure 3 in the main paper showcases a flowerbed and grassy area, which represent scenes with high-frequency textures and complex geometries. Additionally, the sixth row illustrates the truck's window glass — not only are we able to reconstruct the transparent glass surface, but we also faithfully reproduce the reflections on it.** Our method accurately fits these regions and consistently demonstrates superior visual quality compared to the baseline. To further address the reviewer’s concern, we additionally include experiments on the NeRF-Synthetic dataset, which **contains several scenes with strong non-Lambertian effects**. In particular, the Mic, Materials, and Drums scenes exhibit complex view-dependent reflections and specularities. The supplementary experimental results are presented below.
>
> | Scene     | PSNR  | SSIM  | LPIPS  |
> |-----------|-------|-------|--------|
> | mic       | 36.75 | 0.991 | 0.009  |
> | materials | 30.37 | 0.954 | 0.051  |
> | drums     | 26.31 | 0.945 | 0.049  |
>
>
> # Q2. How does the proposed method perform in scenes with large depth ranges or complex multi-layer occlusions, where a single global sorting order may not fully resolve local visibility conflicts?
>
> Our method performs robustly even in scenes with large depth ranges and intricate multi-layer occlusions. It is important to clarify that the "single global sorting" we refer to is a one-time global sorting performed during rendering. Our approach does not maintain a static point cloud; rather, **it dynamically generates neural primitives based on the rendering viewpoint, neural network outputs, and explicit anchors.** In other words, the primitives representing the same scene vary across different viewpoints.
>
> Moreover, our primitives are zero-thickness planar slices aligned with the screen, so **sorting them requires no complex handling of volumetric overlaps.** The occlusion relationships are dynamically determined by the rendering viewpoint and depth sorting during rendering, rather than by fixed geometric topology. This dynamic mechanism enables our method to handle local visibility conflicts effectively, even in challenging scenarios with fine occlusions and large depth variations.
>
> For example, **the fifth row in Figure 3 (the train scene) provides a clear illustration. It contains both distant background elements (the sky) and nearby complex structures (the train)**. If we focus solely on the anchors without considering our rendered primitives, due to the adaptive density control, the anchors in **the sky region are relatively sparse, whereas the foreground train and the gravel road exhibit densely distributed anchors**. These explicit anchors collectively form a coarse geometric structure of the scene.
>
> Each anchor maintains a local neural rendering field. When rendering from a specific viewpoint, these anchors generate corresponding view-dependent planar primitives, which are then globally sorted once to produce correct geometry, appearance, and consistent occlusion relationships. In contrast, prior methods—as illustrated in our Figure 1—**handle overlapping primitives by relying solely on the depth of their centers for ordering.** This approach leads to ambiguous and inconsistent depth relationships across different pixels when primitives overlap.
>
> # Q3. Clarification on the Maximum Gradient Densification Strategy Regarding Noise and High-Frequency Boundaries
>
> This is an excellent question and lies at the core of much ongoing research on 3DGS densification. We attribute the issue to the gradient-based densification methods currently used, which **tend to suffer from gradient vanishing problems in complex scenes**, as noted in works like Pixel-GS and ABS-GS.
> In our method, we observe that in certain challenging scenarios (e.g., noisy regions), **the shapes of rendered primitives vary significantly across different directions.** As demonstrated in Appendix Figure 6(d) and supported by the Pixel-GS paper, the sizes of Gaussians can differ depending on the viewpoint, leading to large discrepancies in gradient magnitudes from different directions. Averaging these gradients often suppresses the high-gradient signals from certain viewpoints, causing densification to fail in those critical regions. **By considering the maximum gradient instead of the average**, our approach effectively mitigates this problem, **enabling more reliable densification in areas with strong directional variation in gradient signals**.
>
> **Balancing Densification and Computational Cost:**
> We present additional experimental results in **Appendix Figure 5** to illustrate this trade-off. As shown, when the number of primitives is relatively low, increasing the number of points correlates almost linearly with improved rendering quality. However, **beyond a certain threshold, further increasing the number of primitives no longer yields noticeable gains in quality.** We further supplement **our analysis with additional comparisons of densification levels in terms of computational cost, memory usage, and training and inference overhead.** We conducted experiments on the train scene of the Tank & Temple dataset, with all tests performed on a single NVIDIA L40S GPU. Here, SAP num denotes the average number of SAP primitives involved in rendering per viewpoint, while anchor refers to the total number of scene anchors.
>
> | PSNR  | SSIM  | LPIPS | SAP num (k) | FPS     | Training Time (min) | Anchor (k) | Memory (MB) |
> |-------|-------|-------|-------------|---------|---------------------|------------|-------------|
> | 23.54 | 0.841 | 0.168 | 1693        | 53.89   | 1530                | 957        | 284.84      |
> | 23.56 | 0.840 | 0.162 | 1020        | 88.20   | 18.95               | 555        | 165.22      |
> | 23.52 | 0.838 | 0.178 | 789         | 98.95   | 16.60               | 392        | 116.74      |
> | 23.31 | 0.828 | 0.188 | 679         | 109.66  | 15.85               | 313        | 93.14       |
> | 23.12 | 0.823 | 0.207 | 588         | 117.00  | 15.95               | 270        | 80.28       |
>
> From the table, it can be observed that increasing the number of primitives significantly improves reconstruction performance when the count is low. However, **beyond a certain threshold, further increases yield diminishing returns.** Regarding **training and inference time costs, these scale roughly proportionally with the average number of SAP primitives per viewpoint**, since the primary computational expense lies in the forward and backward passes during rendering, which are performed per primitive. As for **storage costs, they scale proportionally with the number of anchors**, which also influence the number of SAP primitives generated.
>
> Regarding the trade-off between computational cost and performance, this is often closely tied to hyperparameter choices in 3DGS-based methods. As shown in Appendix Figure 5, **improving the densification strategy can lead to better reconstruction quality under the same computational budget.**
>
>
> >  Zhang, Z.; Hu, W.; Lao, Y.; He, T.; Zhao, H. Pixel GS: Density control with pixel-aware gradient for 3D Gaussian splatting. \
> > Ben Mildenhall, Pratul P Srinivasan, Matthew Tancik, Jonathan T Barron, Ravi Ramamoorthi, and Ren Ng. 2021. Nerf: Representing scenes as neural radiance fields for view synthesis.

---

> > ### Comment · Reviewer_zbg4 · 2025-08-04
> >
> > Thank the authors for their rebuttals, which well-address my questions. While SAP is interesting, I find the idea of primitives changing across different viewpoints less appealing. Nevertheless, my overall assessment remains positive.

---

> > > ### Author Response · Authors · 2025-08-06
> > >
> > > Thank you for your thoughtful follow-up and for maintaining your positive assessment. We truly appreciate your valuable feedback throughout this process. Thank you again for your time and effort.

---

### Note · Authors · 2025-08-13

We express our gratitude to all the reviewers for their valuable insights! Here, we summarize the feedback provided by all the reviewers during the rebuttal period.

**First, we summarize the strengths of our work as highlighted by the reviewers:**

The motivation is intuitive and clear, the method is straightforward, effective, and compatible with existing frameworks. The paper is clear, easy to understand, and easy to follow. The experimental results show improvements over the baseline.

**Next, in response to the concerns raised by the reviewers, our supplementary answers are as follows:**
1. **On ensuring 3D consistency**: We explain that our method does not construct a static model, but instead maintains a dynamic set of viewpoint-dependent primitives for each perspective.
2. **The robustness of sparse viewpoints**: We conducted experiments on standard datasets by reducing the number of training viewpoints at different ratios. The results show that our method still outperforms the baseline.
3. **Expressive power in complex scenes**: We demonstrated that our method outperforms the baseline on highly challenging datasets and pointed out that the qualitative experimental figures in our paper also include these complex scenes. Additionally, we provided supplementary reconstruction results for some challenging scenes.
4. Supplementary **experiments on computational resources**.

**We commit to updating the following in the final camera-ready version:**
1.  We will **add more conceptual descriptions and discussions about multi-view consistency.** Additionally, we will include experiments with reduced viewpoints in Appendix.
2. We will **provide an explanation of how our method operates in complex scenes.**
3. We will remove year-related information from the main tables and **include additional metrics such as time and storage.** More comprehensive experiments on computational resources will be added to Appendix.
4. We will **correct the few minor writing issues currently present.**

Finally, during the rebuttal phase, **all the reviewers expressed that we have addressed their concerns.** We primarily addressed some theoretical issues and further demonstrated the effectiveness of our method through additional experiments. We are confident that our work will contribute to and inspire the community. We would like to once again thank the reviewers and the area chair for their guidance and concern regarding our work.

---

### Decision · Program_Chairs · 2025-09-17

**Decision:**

Accept (poster)

**Comment:**

This paper tackles the depth-sorting problem in 3D Gaussian Splatting, by introducing Screen-Aligned Primitives (SAP), where view-dependent 2D Gaussians aligned with the image plane enable exact and efficient per-pixel sorting. A lightweight decoder generates these primitives from 3D anchors using viewing direction and feature cues. Experiments demonstrate that the proposed method eliminates artifacts while achieving real-time, high-quality rendering.

All reviewers found the paper well-motivated, clearly written, and easy to follow. They acknowledged that the proposed method is simple, effective, and compatible with existing frameworks, while showing consistent improvements over the baseline. However, some concerns were initially raised regarding 3D consistency, robustness under sparse viewpoints, and performance in complex scenes, as well as the completeness of computational evaluations. The authors provided convincing supp. experiments and clarifications in the rebuttal, including viewpoint reduction studies, results on challenging datasets, and additional analyses of computational resources. After the rebuttal, all reviewers agreed that their concerns were sufficiently addressed.

Overall, given the paper’s clarity, clear motivation and demonstrated effectiveness, the AC recommends acceptance.